# Unconstrained Robust Online Convex Optimization

**Jiujia Zhang** [1]   **Ashok Cutkosky** [1]

## Abstract

This paper addresses online learning with "corrupted" feedback. Our learner is provided with potentially corrupted gradients $\tilde{g}_t$ instead of the "true" gradients $g_t$. We make no assumptions about how the corruptions arise: they could be the result of outliers, mislabeled data, or even malicious interference. We focus on the difficult "unconstrained" setting in which our algorithm must maintain low regret with respect to any comparison point $u \in \mathbb{R}^d$. The unconstrained setting is significantly more challenging as existing algorithms suffer extremely high regret even with very tiny amounts of corruption (which is not true in the case of a bounded domain). Our algorithms guarantee regret $\|u\|G(\sqrt{T} + k)$ when $G \geq \max_t \|g_t\|$ is known, where $k$ is a measure of the total amount of corruption. When $G$ is unknown we incur an extra additive penalty of $(\|u\|^2 + G^2)k$.

## 1. Introduction

In this paper, we consider unconstrained online convex optimization (OCO) under the presence of adversarial corruptions. In general, OCO is a framework in which a learner iteratively outputs a prediction $w_t \in \mathcal{W}$, then observes a vector $g_t = \nabla \ell_t(w_t)$ for some convex loss function $\ell_t : \mathcal{W} \to \mathbb{R}$, and then incurs a loss of $\ell_t(w_t)$. The learner's performance over a time horizon $T$ is evaluated by the *regret* relative to a fixed competitor $u \in \mathcal{W}$, denoted as $R_T(u)$:

$$R_T(u) := \sum_{t=1}^{T} \langle g_t, w_t - u \rangle \geq \sum_{t=1}^{T} \ell_t(w_t) - \ell_t(u)$$

The inequality above follows by convexity of $\ell_t$. Classical results in this field consider a bounded domain $\mathcal{W}$ with

[1]Department of Electrical and Computer Engineering, Boston University, Boston, USA. Correspondence to: Jiujia Zhang <jiujiaz@bu.edu>, Ashok Cutkosky <ashok@cutkosky.com>.

*Proceedings of the 42$^{st}$ International Conference on Machine Learning*, Vancouver, Canada. PMLR 267, 2025. Copyright 2025 by the author(s).

known diameter $D$ and a Lipschitz bound $G \geq \max_t \|g_t\|$. In this setting, the optimal bound is $R_T(u) \leq O(GD\sqrt{T})$ (Zinkevich, 2003; Abernethy et al., 2008).

Our work focuses on the *unconstrained* case $\mathcal{W} = \mathbb{R}^d$, where it is typical to aim for a regret guarantee that scales not with a uniform diameter bound $D$, but with the norm of the comparator $\|u\|$. Such bounds are often called "comparator adaptive" (because they adapt to the comparator $u$), or "parameter-free" (because this adaptivity suggests that the algorithms require less hyperparameter tuning). In this unconstrained setting, the classical algorithms achieve $R_T(u) = \tilde{O}(\|u\|G\sqrt{T})$ (Mcmahan & Streeter, 2012; McMahan & Orabona, 2014; Orabona & Pál, 2016; Orabona, 2014) (which is also optimal).

We are interested in a harder variant of the OCO framework with "corrupted" gradients. Specifically, instead of any direct information about the function $\ell_t$, after each round the learner is provided with a vector $\tilde{g}_t$ that should be interpreted as an estimate of $g_t = \nabla \ell_t(w_t)$. Our aim is to obtain a regret that scales as $\|u\|G(\sqrt{T} + k)$ for all $u \in \mathcal{W}$, where $k$ is some measure of the degree to which $\tilde{g}_t \neq g_t$ that will be formally defined in Section 2. Roughly speaking, $k$ can be interpreted as the number of rounds in which $\tilde{g}_t \neq g_t$. Notably, the desired rate is robust to adversarial corruptions in the sense that it allows $k = O(\sqrt{T})$ before the bound becomes worse than the optimal result *without* corruptions.

Our dual challenges of corrupted $\tilde{g}_t$ and unconstrained $\mathcal{W}$ are naturally motivated by problems in practice. The unconstrained setting is ubiquitous in machine learning - consider the classical logistic regression setting, for which it is unusual to impose constraints. The corrupted $\tilde{g}_t$ in contrast is less commonly studied, but represents a common practical issue: the computed gradients may not be good estimates of a "true" gradient, either due to the presence of statistical outliers, numerical precision issues in the gradient computation, or mislabeled or otherwise damaged data.

We distinguish two different settings in our results: one in which the algorithm is provided with prior knowledge of a number $G \geq \max_t \|g_t\|$, and one in which it is not. This is a common dichotomy in unconstrained OCO, even without corruptions. In the former case, the classical result of $\tilde{O}(\|u\|G\sqrt{T})$ is obtainable, while in the latter case it is not: instead the optimal results are $R_T(u) \leq$

$\tilde{O}(\|u\| \max_t \|g_t\| \sqrt{T} + \|u\|^3 \max_t \|g_t\|)$ (Cutkosky, 2019; Mhammedi & Koolen, 2020), or $\tilde{O}(\|u\| \max_t \|g_t\| \sqrt{T} + \|u\|^2 + \max_t \|g_t\|^2)$ by Cutkosky & Mhammedi (2024). The later excels particularly whenever $G$ is not excessively large: $G \le \|u\| \sqrt{T}$.

To the best of our knowledge, the setting of unconstrained OCO with corruptions has not been studied before. Perhaps the closest works to ours are Zhang & Cutkosky (2022); Jun & Orabona (2019); van der Hoeven (2019) and van Erven et al. (2021). (Zhang & Cutkosky, 2022; Jun & Orabona, 2019; van der Hoeven, 2019) study the unconstrained setting, but assume that $\tilde{g}_t$ is a random value with $\mathbb{E}[\tilde{g}_t] = g_t$. In contrast, we assume no such stochastic structure on $\tilde{g}_t$. On the other hand, van Erven et al. (2021) does not make any assumptions about the nature of the corruptions, but assumes that $\mathcal{W}$ has finite diameter $D$. They achieve a regret of $O(DG(\sqrt{T} + k))$. Our development will borrow some ideas from van Erven et al. (2021) with the aim to bound $R_T(u)$, but we face unique difficulties. As detailed in Section 3, the unconstrained setting means that even very small corruptions could have dire consequences (unlike in the constrained setting). Moreover, if the Lipschitz constant $G$ is not known, the problem becomes even more challenging; as detailed in Section 6.1, prior methods for handling unknown $G$ do not apply because we never learn $G$ even at the end of all $T$ rounds.

The notion of adversarial corruption is common in the field of robust statistics, with early efforts focusing primarily on the presence of outliers in linear regression (Huber, 2004; Cook, 2000; Thode, 2002). These inspired broader application in machine learning, asuch as Robust PCA (Candès et al., 2011), anomaly detection (Raginsky et al., 2012; Delibalta et al., 2016; Zhou & Paffenroth, 2017; Sankararaman et al., 2022), robust regression (Klivans et al., 2018; Cherapanamjeri et al., 2020; Chen et al., 2022), and mean estimation (Lugosi & Mendelson, 2021). For a comprehensive review of recent advances in this area, see Diakonikolas & Kane (2019).

Adversarial corruption has also been studied in iterative optimization setting other than OCO, such as in stochastic bandits (Lykouris et al., 2018; Gupta et al., 2019; Ito, 2021; Agarwal et al., 2021) and stochastic optimization (Chang et al., 2022; Sankararaman & Narayanaswamy, 2024).

**Contributions and Organization**  In the case that the algorithm is given prior knowledge of $G$, we provide an algorithm that achieves $R_T(u) = \tilde{O}(\|u\| G(\sqrt{T} + k))$ in Section 5.1, with a matching lower bound (see Section 5.2). Alternatively, when $G$ is unknown, a regret bound with an additional penalty of $(\|u\|^2 + G^2)k$ is attained (see Section 6.3).

Meanwhile, we provide two specific applications of our re-

sults in Appendix D. First, we show that our method can be used to solve stochastic convex optimization problems in some of the gradient computations are altered in an arbitrary way. Second, we solve a natural "online" version of a distributionally robust optimization problem. Before providing our main results, we introduce notation and define our corruption model in Section 2.

## 2. Notation and Problem Setup

**Notation**  For each $t$, $\ell_t : \mathcal{W} \to \mathbb{R}$ is a convex function, where and $\mathcal{W} = \mathbb{R}^d$. Let $w_t \in \mathcal{W}$ be iterates from some online learning algorithm and denote $g_t = \nabla \ell_t(w_t)$ as the "true" (sub)gradient. Let $\tilde{g}_t$ be the the corrupted gradient observed by the learner. Define $\mathbb{1}\{\cdot\}$ as the indicator function, where $\mathbb{1}\{\text{TRUE}\} = 1, \mathbb{1}\{\text{FALSE}\} = 0$. We use $|\cdot|$ to denote the cardinality of a set. Let $\|\cdot\|$ denote the Euclidean norm. Denote $\mathbb{R}^+ = \{x \in \mathbb{R} : x \ge 0\}$. We define shorthand notation for sets $[T] = \{1, 2, \dots, T\}$ and $[a, T] = \{a, a+1, \dots, T\}$ for any $a \in [T]$. We use $\mathcal{B} \subseteq [T]$ to denote an index set, and $\bar{\mathcal{B}} = [T] \setminus \mathcal{B}$ for its complement. We use $O(\cdot)$ to hide constant factors and $\tilde{O}(\cdot)$ to additionally conceal any polylogarithmic factors.

**Problem Setup**  Instead of the true gradients $g_t$, our algorithms only receive potentially corrupted gradients $\tilde{g}_t$. Two natural measures to quantify corruptions are:

$$k_{\text{count}} := \sum_{t=1}^{T} \mathbb{1}\{g_t \ne \tilde{g}_t\} \tag{1}$$

$$k_{\text{deviation}} := \frac{1}{G} \sum_{t=1}^{T} \|g_t - \tilde{g}_t\| \tag{2}$$

where $G$ is a scalar that satisfies $G \ge \max_t \|g_t\|$ and is often referred as the "Lipschitz constant". The metric $k_{\text{count}}$ counts the rounds in which $\tilde{g}_t \ne g_t$ but allowing for arbitrarily large deviations $\|\tilde{g}_t - g_t\|$ in those rounds. This is suitable for detecting outlier effects and highlighted in studies such as (van Erven et al., 2021; Sankararaman & Narayanaswamy, 2024). Conversely, $k_{\text{deviation}}$ measures the cumulative deviation, accommodating corruption in every round, making it optimal for identifying subtle yet widespread errors or malicious activities, akin to the issues addressed in (Lykouris et al., 2018; Gupta et al., 2019; Ito, 2021; Agarwal et al., 2021; Chang et al., 2022).

In order to provide a unified way to study those two distinct corruption measures in Equation (1) and (2), we assume that our algorithm is provided with a number $k$ that satisfies:

$$|\mathcal{B}| := |\{t \in [T] : \|g_t - \tilde{g}_t\| \ge G\}| \le k \tag{3}$$

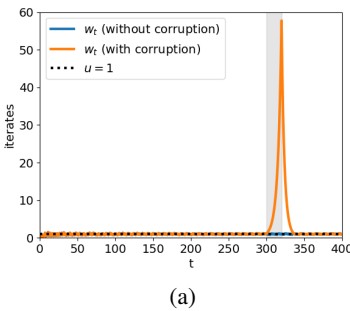 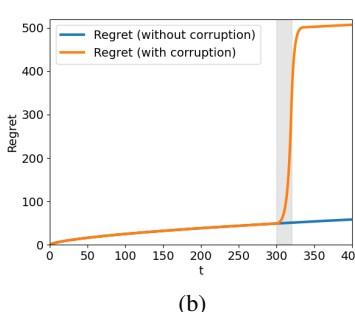 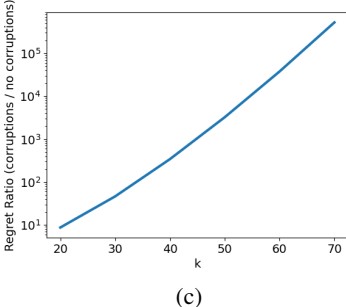

(a)       (b)       (c)

*Figure 1.* KT-bettor with $\ell(w) = |w - 1|$ and comparator $u = 1$. (a)-(b): $T = 400$ and corruption happens during $t \in [300, 319]$. (c): Ratio between regrets with and without corruptions with various total corrupted rounds $k \in [20, 30, 40, 50, 60, 70]$ and $T = k^2$.

and also:

$$\frac{1}{G} \sum_{t=1}^{T} \min\left(\|g_t - \tilde{g}_t\|, G\right) \leq k \qquad (4)$$

Intuitively, $\mathcal{B}$ denotes the rounds with a "large" amount of corruption. Notice that

$$|\mathcal{B}| \leq \min\left(k_{\text{count}}, k_{\text{deviation}}\right)$$

and

$$\frac{1}{G} \sum_{t=1}^{T} \min\left(\|g_t - \tilde{g}_t\|, G\right) \leq \min\left(k_{\text{count}}, k_{\text{deviation}}\right)$$

Hence, an algorithm whose complexity depends on $k$ satisfying Equation (3) and (4) can be used with either $k = k_{\text{count}}$ or $k = k_{\text{deviation}}$ depending on which is more appropriate to the problem at hand.

## 3. Challenges in Unconstrained Domain

Dealing with corruptions with an unconstrained domain is significantly more challenging than one with a bounded domain. As an illustration of the added difficulty, suppose the corruptions are so "small" that $\|g_t - \tilde{g}_t\| \leq G$ for all $t$. Then in a bounded $\mathcal{W}$ with a diameter $D$, an algorithm that completely ignores the possibility of corruptions and directly runs on $\tilde{g}_t$ will have low regret. This can be seen as follows: since $\|u - w_t\| \leq D$ for every $u, w_t \in \mathcal{W}$, we have:

$$\sum_{t=1}^{T} \langle g_t, w_t - u \rangle \leq \sum_{t=1}^{T} \langle \tilde{g}_t, w_t - u \rangle + \sum_{t=1}^{T} \|g_t - \tilde{g}_t\| \|w_t - u\|$$
$$\leq \sum_{t=1}^{T} \langle \tilde{g}_t, w_t - u \rangle + kGD$$

In this case, $\|u - w_t\| \leq D$ prevents the algorithm from straying too far from the comparator $u$.

The situation is much more difficult in the *unconstrained* setting. Algorithm for this setting typically produce outputs $w_t$ that potentially grow *exponentially fast* in order to quickly compete with comparators that are very far from the starting point. However, this also means the algorithm is especially fragile to corruption since the growth of $w_t$ can be highly sensitive to deviations in $\|g_t - \tilde{g}_t\|$. Even a small deviation could cause $w_t$ to move extremely far away and therefore incur a very high regret. This phenomenon is illustrated in Figure 1 with the KT-bettor algorithm (Orabona & Pál, 2016), which is a standard example of an unconstrained learner.

In Figure 1, we considered $\ell_t(w) = |x - 1|$ for all $t$. Figure 1a and 1b demonstrate $k = 20$ gradients being corrupted by setting $\tilde{g}_t = -g_t$ during rounds $t \in [300, 300 + k - 1] = [300, 319]$ over a time span of $T = k^2 = 400$. This results in an exponential deviation away from the comparator $u = 1$ and so incurs a high regret. Finally, we show that this problem becomes exacerbated as $k$ increases by simulating $k \in [20, 30, 40, 50, 60, 70]$ for $T = k^2$ in Figure 1c.

## 4. Robustification through Regularization

In this section, we outline our general algorithm-design recipe. When receiving possibly corrupted gradients $\tilde{g}_t$, we first employ a gradient clipping step with some threshold $h_t$ that outputs a "clipped" version $\tilde{g}_t^c$, defined as follows:

$$\tilde{g}_t^c = \frac{\tilde{g}_t}{\|\tilde{g}_t\|} \min\left(h_t, \|\tilde{g}_t\|\right) \qquad (5)$$

This preprocessing step "corrects" some corruption effect when $h_t$ is appropriately chosen. For example, in the case of $h_t = G \geq \max_t \|g_t\|$, then $\tilde{g}_t^c$ is always "less corrupted" than $\tilde{g}_t$, as $\|\tilde{g}_t^c - g_t\| \leq \|\tilde{g}_t - g_t\|$. Then $\tilde{g}_t^c$ is used as a feedback to an online learner, yielding the following expression

for $R_T(u)$:

$$R_T(u) := \sum_{t=1}^{T} \langle g_t, w_t - u \rangle$$

$$= \sum_{t=1}^{T} \langle \tilde{g}_t^c, w_t - u \rangle + \sum_{t=1}^{T} \langle g_t - \tilde{g}_t^c, w_t - u \rangle$$

Next, we incorporate a regularization function $r_t : \mathcal{W} \to \mathbb{R}$. We can re-write $R_T(u)$ as the following:

$$R_T(u) := \underbrace{\sum_{t=1}^{T} \langle \tilde{g}_t^c, w_t - u \rangle + r_t(w_t) - r_t(u)}_{:= R_T^{\mathcal{A}}(u)}$$

$$+ \underbrace{\sum_{t=1}^{T} \langle g_t - \tilde{g}_t^c, w_t \rangle}_{:= \text{ERROR}} - \underbrace{\sum_{t=1}^{T} r_t(w_t)}_{:= \text{CORRECTION}}$$

$$+ \underbrace{\left\langle \sum_{t=1}^{T} g_t - \tilde{g}_t^c, -u \right\rangle + \sum_{t=1}^{T} r_t(u)}_{:= \text{BIAS}} \quad (6)$$

The interpretation is that $R_T(u)$ can be controlled through four components ERROR, CORRECTION, BIAS and a composite regret $R_T^{\mathcal{A}}(u)$. Here, $R_T^{\mathcal{A}}(u)$ is the regret of an online learner $\mathcal{A}$ with respect to the losses $\langle \tilde{g}_t^c, w \rangle + r_t(w)$. Some information about the structure of $r_t$ may be known *before* $w_t$ is chosen, making this problem similar to online optimization with *composite* losses (Duchi et al., 2010). We summarize the general procedure as Algorithm 1. Depending on whether $G \geq \max_t \|g_t\|$ is known or not, the appropriate settings for $h_t$, the selection of regularizer $r_t$ and the online learner $\mathcal{A}$ differs. We apply this general framework to the case where $G$ is known in Section 5.1 first, and the consider the significantly more challenging unknown-$G$ setting in Section 6.

---

**Algorithm 1** General Protocol

---

1: **Input:** Clipping thresholds: $0 < h_1 \leq \cdots \leq h_{T+1}$
   An online learning algorithms $\mathcal{A}$.
   A regularizer: $r_t : \mathcal{W} \to \mathbb{R}^+$
2: **for** $t = 1$ **to** $T$ **do**
3:     Play $w_t$, receive $\tilde{g}_t$
4:     Compute $\tilde{g}_t^c$ as Eqn. (5)
5:     Send $\tilde{g}_t^c, h_{t+1}$ to $\mathcal{A}$ and get $w_{t+1}$
6: **end for**

---

## 5. Robust Learning with Knowledge of Lipschitz Constant

In this section, we proceed under the simpler setting that $G \geq \max_t \|g_t\|$ is known a priori. We therefore will set

$h_t = G$ for all iterations in the definition of $\tilde{g}_t^c$ (see Equation (5)).

### 5.1. The Algorithm and Regret Guarantee

The quantity ERROR as defined in Equation (6) can be upper bounded by:

$$\text{ERROR} \leq \left( \sum_{t \in \mathcal{B}} \|g_t - \tilde{g}_t^c\| + \sum_{t \in \bar{\mathcal{B}}} \|g_t - \tilde{g}_t^c\| \right) \max_t \|w_t\|$$

$$\leq \left( \sum_{t \in \mathcal{B}} \|g_t - \tilde{g}_t\| + G|\bar{\mathcal{B}}| \right) \max_t \|w_t\|$$

$$\leq kG \max_t \|w_t\|$$

where $\mathcal{B}$ is defined in Equation (3), and $\bar{\mathcal{B}} = [T] \setminus \mathcal{B}$. The second line is due to $\|g_t - \tilde{g}_t^c\| \leq \|g_t - \tilde{g}_t\| \leq G, \forall t \in \bar{\mathcal{B}}$, because $h_t = G \geq \|g_t\|$. The last inequality is due to the corruption model presented in Equation (4). This suggests that the worst case value for ERROR is at most $kG \max_t \|w_t\|$. This could potentially be exponential in $t$ as shown in Lemma 8 (Zhang & Cutkosky, 2022). On the other hand, no matter which regularizer $r_t$ we choose, a worst case upper bound on BIAS can be derived:

$$\text{BIAS} \leq kG\|u\| + \sum_{t=1}^{T} r_t(u)$$

To enable CORRECTION to cancel ERROR while ensuring BIAS does not grow too large, the ideal choice of regularizer $r_t$ should achieve the following bounds simultaneously:

$$\sum_{t=1}^{T} r_t(w_t) \geq \tilde{\Omega}(kG \max_t \|w_t\|), \quad \sum_{t=1}^{T} r_t(u) \leq \tilde{O}(kG\|u\|)$$

To accomplish this, we choose $r_t(w) = f_t(w)$ as displayed in Equation (7), which belongs to family of Huber losses first proposed by (Zhang & Cutkosky, 2022):

$$f_t(w; c, p, \alpha) = c\sigma_t(w; p, \alpha)/S_t^{1-1/p} \quad (7)$$

where

$$S_t = \sum_{i=1}^{t} \|w_i\|^p + \alpha^p \quad (8)$$

and a piecewise function $\sigma_t$:

$$\sigma_t(w; p, \alpha) = \begin{cases} \|w\|^p, & \|w\| \leq \|w_t\| \\ (p\|w\| - (p-1)\|w_t\|)\|w_t\|^{p-1}, & \text{otherwise} \end{cases} \quad (9)$$

This function behaves polynomially near $\|w_t\|$ and linear otherwise. By setting $c = kG, p = \ln T, \alpha = \epsilon/k$, $f_t$

achieves the desired properties. See a detailed discussion of the characteristics of this family of losses in (Zhang & Cutkosky, 2022). For completeness we include a proof of the relevant properties in Lemma B.1. Overall, these bounds allow ERROR − CORRECTION $\leq \tilde{O}(1)$ and BIAS $\leq \tilde{O}(kG\|u\|)$.

Armed with this peculiar regularization function, we now need an online learner that can control $R_T^A(u)$. This is similar to online learning with loss $\tilde{\ell}_t(w_t) := \langle \tilde{g}_t^c, w_t \rangle + r_t(w_t)$ at each round $t$. However, we cannot simply apply any standard OCO algorithm to these losses. The issue is that regularizer $r_t$ is $\Theta(kG)$ Lipschitz rather than $G$-Lipschitz. So, a naive approach would result in $R_T^A(u) = \tilde{O}(\|u\|kG\sqrt{T})$, but we want our final regret to be only $\tilde{O}(\|u\|G(\sqrt{T}+k))$. The key to avoid the multiplicative $k$-factor is to observe that $r_t$ is known ahead-of-time: it is a *composite* term in the loss, and so ideally the regret should only depend on $\tilde{g}_t^c$. Unfortunately, composite losses are not as well-studied in the unconstrained online learning literature. Moreover, our composite loss is somewhat non-standard in that the shape of the function $r_t$ depends slightly on $w_t$. Nevertheless, we show that in fact the online mirror descent algorithm developed by (Jacobsen & Cutkosky, 2022) actually does guarantee composite regret in our setting. That is $R_T^A(u) \leq \tilde{O}(\|u\|G\sqrt{T})$ (see Theorem C.2, and an explicit algorithmic update procedure in Algorithm 2). By combining these ingredients, we are ready to state the overall regret bound, whose proof is deferred to Appendix D.

**Theorem 5.1.** *Suppose $g_t, \tilde{g}_t$ satisfies assumptions in Equation (3) and (4). Setting $h_t = G, r_t = f_t$ as defined in Equation (7) with $c = kG, p = \ln T, \alpha = \epsilon/k$ for some $\epsilon > 0$, set Algorithm 2 algorithm as the base algorithm $A$. Then Algorithm 1 guarantees:*

$$R_T(u) \leq \tilde{O}\left[\epsilon G + \|u\|G\left(\sqrt{T}+k\right)\right]$$

Theorem 5.1 shows that the penalty for corrupted gradients is at most $\tilde{O}(\|u\|Gk)$. This result has a few intriguing properties. First, so long as $k \leq \sqrt{T}$, the penalty is subasymptotic to the standard uncorrupted regret bound $\tilde{O}(\|u\|G\sqrt{T})$. That is, we can tolerate $k$ up to $\sqrt{T}$ essentially "for free". Next, observe that for $u = 0$, the regret is $\epsilon$ no matter what $k$ is. Constant regret at the origin is typical for unconstrained algorithms, but is especially remarkable for our corrupted setting. Imagine a scenario in which we define $0$ to represent some "default" action. Our bound then suggests that *no matter how much corruption is present*, we never do significantly worse than this default.

### 5.2. Lower Bounds

We present a lower bound in Theorem 5.2 with proofs deferred in Appendix E. This result shows that the upper bound

of Theorem 5.1 is tight with respect to some comparator $u^*$ with any pre-specified magnitude. In addition, we provide a second lower bound as Theorem E.5 in Appendix E, which has the matching log factor.

**Theorem 5.2.** *For every $D > 0$, there exists a comparator $u^* \in \mathbb{R}^d$ such that $\|u^*\| = D$, $\tilde{g}_1, \cdots, \tilde{g}_T$ and $g_1, \cdots, g_T$ such that $\|g_t\|, \|\tilde{g}_t\| \leq 1$, $\sum_{t=1}^T \mathbb{1}\{\tilde{g}_t \neq g_t\} = k$:*

$$\sum_{t=1}^T \langle g_t, w_t - u^* \rangle \geq \Omega\left[\|u^*\|\left(\sqrt{T}+k\right)\right]$$

## 6. Robust Learning with Unknown Lipschitz Constant

In this section, we consider the scenario where $G \geq \max_t \|g_t\|$ is unknown. We first discuss the additional challenges must be addressed in Section 6.1, followed by intuition of selecting $h_t, r_t$. Finally, we discuss the choice of the base algorithm $A$ and the regret guarantee.

### 6.1. Challenges with Unknown $G$

The combination of unconstrained domain, unknown $G$ and corrupted gradients provides a set of challenges that stymie current techniques in the literature. Let us unpack these challenges carefully, starting from where our analysis in Section 5 breaks down.

Previously, we employed a clipping threshold $h_t = G$ to automatically filter out $\tilde{g}_t$ values that were in some sense "obviously corrupted", replacing them with values that were only corrupted by at most $G$. Then, we employed a technical regularization scheme by setting $r_t = f_t$ to "cancel" out these bounded corruptions. This strategy is no longer available: we do not know $G$.

A natural approach would be to maintain a time-varying threshold $h_t$ that estimates $G$ on-the-fly, and use it in place of the unknown constant $G$. However, this is harder than it seems because inevitably we will sometimes have $h_t < G$, and so we are likely to clip even an uncorrupted gradient: $\tilde{g}_t^c \neq g_t$ even when $\tilde{g}_t = g_t$. This means that unlike the known $G$-case, our clipping operation is not purely benign. Nevertheless, methods in the literature for tackling the unknown-$G$ setting *without corruptions* often use exactly this method with $h_t = \max_{i<t}\|g_i\| \leq G$ (Cutkosky, 2019; Mhammedi & Koolen, 2020; Cutkosky & Mhammedi, 2024).

Unfortunately, this does not work for our corrupted setting. The new difficulty is that we *never know the value $G$, even in hindsight*. The choice $h_t = \max_{i<t}\|\tilde{g}_i\|$ is bad: a single corrupted gradient with $\|\tilde{g}_t\| \gg G$ will be catastrophic. Moreover, even if we were guaranteed that $\|\tilde{g}_t\| \leq G$ for all $t$, it might be that $\|g_t\| \geq h_t$, so that truncation actually

results in $\|g_t - \tilde{g}_t^c\| \geq \|g_t - \tilde{g}_t\|$. We will call this additional bias a "truncation error". As a result, we need to address a new challenge:

1. How can we choose $h_t$ in a robust way that serves as a good estimate of $G$ without introducing too much truncation error?

In addition, even if we could address challenge 1 and have access to $h_t \lesssim G$, the regularizer $r_t = f_t$ used for our known-$G$ algorithm *also* required exact knowledge of the value for $G$. We rely on the scaling factor $c \geq O(kG)$ to be big enough to "cancel" the ERROR $\leq kG \max_t \|w_t\|$ as $\sum_t f_t(w_t) \geq \tilde{O}(c \max_t \|w_t\|)$. Therefore, simply setting $c = kh_t$ is not sufficient since $h_t$ values are likely smaller than $G$.

In the known-$G$ case, the Huber regularizer $f_t$ almost perfectly cancels the error by imposing a penalty that is also roughly proportional to $\max_t \|w_t\|$. The challenge in the unknown $G$-case is that the proportionality constant of $kG$ is unknown, so we cannot perform this cancellation. Instead, we propose to use a regularizer $r_t(w) = O(\|w\|^2)$ that grows faster than the Huber regularizer $f_t$ as illustrated in Figure 2. This way, $kG \max_t \|w_t\| \leq r_t(\max_t \|w_t\|)$ for large enough $\|w_t\|$ no matter what $G$ is. By scaling $r_t$ appropriately, we would like to ensure $\sum_t r_t(w_t) \geq \Omega(k \max_t \|w_t\|^2)$. Then, as long as $\max_t \|w_t\| \geq G$, $\sum_t r_t(w_t)$ will completely cancel ERROR. So overall, without prior knowledge of $G$, ERROR $-$ CORRECTION $\leq O(kG^2)$ is guaranteed. However, the quadratic growth of $r_t$ introduces another challenge:

2. How can we exploit a more aggressive quadratic regularization without introducing too much regularization bias $\sum_t r_t(u)$ for arbitrary $u \in \mathcal{W}$?

In particular, notice that any *constant* regularizer $r_t(u) = C\|w\|^2$ would add bias of $\sum_t r_t(u) = TC\|u\|^2$ to the regret, so that we would require $C = O(1/T)$. This in turn is too small to ensure $\sum_t r_t(w_t) \geq \Omega(k \max_t \|w_t\|^2)$. Our solution will require a more nuanced time-varying regularizer.

### 6.2. Adaptive Thresholding and Error Correction

**Challenge 1:** To meet the challenge of choosing $h_t$ properly, we introduce a "tracking mechanism" that conservatively increases $h_t$ over time. The key insight is that at most $k$ values of $t$ can have $\|\tilde{g}_t\| > 2G$ (See Lemma F.1). Based on this observation, we propose a simple way to maintain a "threshold" $h_t$ which provides a conservative lower bound estimate of $G$. The threshold $h_t$ starts with some small value $h_1 = \tau_G > 0$ and stays constant until we observe $k + 1$

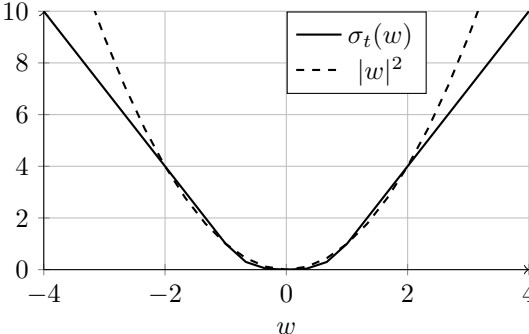

*Figure 2.* Comparison of Huber loss $f_t$ without scaling factor $\sigma_t$ with $p = 3$, $|w_t| = 1$, and $|w|^2$. $|w|^2$ always grow faster than $\sigma_t(w)$ far away from the origin.

iterations in which $\|\tilde{g}_t\| \geq h_t$. This mechanism is named as FILTER and is displayed as Algorithm 3 in Appendix F.

Notice that $h_t$ only doubles if it is guaranteed that some $g_t$ satisfies $h_t \leq \|g_t\|$, so that $h_t \leq O(G)$ always. Denote rounds where gradients are clipped as $\bar{\mathcal{P}} = \{t \in [T] : \tilde{g}_t \neq \tilde{g}_t^c\}$, that is $\bar{\mathcal{P}}$ denotes rounds that have $h_t \leq \|g_t\|$. The conditional doubling mechanism also ensures $|\bar{\mathcal{P}}| \leq \tilde{O}(k)$ (See Lemma F.2). This means only a small fraction of $\tilde{g}_t$ are truncated before $h_t$ becomes a very accurate estimate of $G$. Hence, the total rounds which are vulnerable to suffering from truncation error is small. This FILTER strategy improves upon a method with a similar purpose in van Erven et al. (2021); it uses only constant space rather than $O(k)$ space.

**Challenge 2:** As a consequence of using $h_t$ from FILTER as the clipping threshold, we can decompose the ERROR term defined in Equation (6) by using $\tilde{g}_t = \tilde{g}_t^c$ for $t \in \mathcal{P}$:

$$\text{ERROR} \leq \underbrace{\sum_{t \in \bar{\mathcal{P}}} \|g_t - \tilde{g}_t^c\| \|w_t\|}_{E_{\bar{\mathcal{P}}}: \text{truncation error}} + \underbrace{\sum_{t \in \mathcal{P}} \|g_t - \tilde{g}_t\| \|w_t\|}_{E_{\mathcal{P}}: \text{corruption error}} \quad (10)$$

Therefore, we must choose a regularizer $r_t$ to offset those errors. We discuss how each error component is treated and finally reveal the structure of $r_t$.

Truncation error $E_{\bar{\mathcal{P}}}$ can be reduced by keeping $\|w_t\|$ from being too large. Thus, every time we experience some truncation error, we would like to use some regularization to force the learner to decrease $\|w_t\|$. To this end, we use a quadratic regularization penalty $\alpha_t\|w_t\|^2$ where $\alpha_t \neq 0$ only when $t \in \bar{\mathcal{P}}$ so as to only encourage $w_t$ to decrease only when we have "evidence" that truncation error has occurred. Then, for any round $t \in \bar{\mathcal{P}}$ we bound the truncation error

minus the quadratic regularizer $\alpha_t \|w_t\|^2$ as follows:

$$\|g_t - \tilde{g}_t^c\|\|w_t\| - \alpha_t\|w_t\|^2 \leq$$

$$\sup_{X \geq 0} \|g_t - \tilde{g}_t^c\|X - \alpha_t X^2 = \frac{\|g_t - \tilde{g}_t^c\|^2}{4\alpha_t}$$

such a regularization scheme implies overall:

$$E_{\bar{\mathcal{P}}} - \sum_t \alpha_t\|w_t\|^2 \leq \sum_{t \in \bar{\mathcal{P}}} \frac{\|g_t - \tilde{g}_t^c\|^2}{4\alpha_t} \leq G^2 \sum_{t \in \bar{\mathcal{P}}} \frac{1}{\alpha_t}$$

and additional regularization bias is $\|u\|^2 \sum_t \alpha_t$. This means the truncation error will be controlled as $O(kG^2)$, and the additional bias from this regularization will also be mild as $O(k\|u\|^2)$, if the following conditions hold simultaneously:

$$\sum_{t \in \bar{\mathcal{P}}} 1/\alpha_t \leq O(k), \qquad \sum_{t \in [T]} \alpha_t \leq O(k)$$

Notice that $|\bar{\mathcal{P}}| \leq \tilde{O}(k)$ and that $h_{t+1} \neq h_t, \forall t \in \bar{\mathcal{P}}$. Thus, the following $\alpha_t$ fulfills both requirements:

$$\alpha_t = \gamma_\alpha \cdot \mathbb{1}\{h_{t+1} \neq h_t\} \tag{11}$$

by setting $\gamma_\alpha = O(1)$.

Next, for managing the corruption error $E_{\mathcal{P}}$, one might attempt to take the same approach by imposing an additional quadratic regularizer $\beta_t\|w_t\|^2$. Similar calculation as used in the control of truncation error suggests that we can control the corruption error so long as $\sum_{t \in \mathcal{P}} 1/\beta_t \leq O(k)$ and $\sum_t \beta_t \leq O(k)$ holds simultaneously. Unfortunately, $|\mathcal{P}|$ could potentially be as large as $\Omega(T)$, and so it is not possible to pick constant $\beta_t, \forall t \in \mathcal{P}$ that satisfies these desired rates.

Our remedy is to apply a milder regularization by combining the quadratic regularizer which is active only on a smaller subset $\mathcal{P}_0 \subseteq \mathcal{P}$ and the Huber regularization $f_t$ which is active at every round. To see how this will work, suppose we divide the $T$ rounds into $N$ "epochs" in which $\max_t \|w_t\| \in [2^N, 2^{N+1}]$. Specifically, we begin with some $z_1 = \tau_D > 0$. Whenever $\|w_t\| \geq z_t$, we set $z_{t+1} = 2\|w_t\|$, and $z_{t+1} = z_t$ otherwise (formally summarized as TRACKER as Algorithm 4, Appendix G). We choose $\mathcal{P}_0 = \{t : z_{t+1} \neq z_t\}$, that is time steps $\mathcal{P}_0 = \{\tau_1, \ldots, \tau_N\}$ when threshold doubles. For any two consecutive time steps $\tau_n, \tau_{n+1} \in \mathcal{P}_0$, we define $[\tau_n : \tau_{n+1} - 1]$ as a single "epoch" for each $n \in N$. Notice that for $t \in [\tau_n, \tau_{n+1} - 1]$, we must have $\|w_t\| \leq 2\|w_{\tau_n}\|$. Further, the total corruption error incurred in one such "epoch" is:

$$\sum_{t=\tau_n}^{\tau_{n+1}-1} \|g_t - \tilde{g}_t^c\|\|w_t\| \leq 2\|w_{\tau_n}\| \sum_{t=\tau_n}^{\tau_{n+1}-1} \|g_t - \tilde{g}_t^c\|$$

$$\leq 2kG\|w_{\tau_n}\|$$

So, applying a sufficiently large regularization on exactly the indices in $\mathcal{P}_0$ will ideally be enough to cancel all of the corruption error. Further, notice that $|\mathcal{P}_0| \leq O(\ln(\max_t \|w_t\|))$. Thus, one might hope to apply an approach similar to the one used for truncation error: use a regularizer $\beta_t\|w\|^2$ with $\beta_t = 0$ for $t \notin \mathcal{P}_0$ and $\beta_t$ equal to some constant $\beta$ for $t \in \mathcal{P}_0$. Unfortunately, this will yield a regularization bias of $\sum_{t \in \mathcal{P}_0} \beta\|u\|^2 \leq O(\beta \ln(\max_t \|w_t\|)\|u\|^2)$. While this seems benign, recall that actually $\max_t \|w_t\|$ may be *exponential* in $T$, so that $\ln(\max_t \|w_t\|)$ is still polynomial in $T$. To resolve this, we gradually attenuate $\beta_t$ by choosing:

$$\beta_t = \gamma_\beta \cdot \frac{\mathbb{1}\{z_{t+1} \neq z_t\}}{1 + \sum_{i=1}^t \mathbb{1}\{z_{i+1} \neq z_i\}} \tag{12}$$

With this choice, the following are guaranteed:

$$\sum_{t \in \mathcal{P}_0} 1/\beta_t \leq \tilde{O}\left(\frac{1}{k} \ln \max_t \|w_t\|\right)$$

and

$$\sum_{t \in [T]} \beta_t = \tilde{O}(k \ln \ln \max_t \|w_t\|) = \tilde{O}(k)$$

with $\gamma_\beta = O(k)$. The first identity guarantees that $E_{\mathcal{P}} - \sum_t \beta_t\|w_t\|^2$ is at most $\tilde{O}(kG^2 \ln \max_t \|w_t\|)$, while the second ensures that the additional bias is only $\sum_{t=1}^T \beta_t\|u\|^2 \leq \tilde{O}(k\|u\|^2)$. Now, we still need to handle the potentially-large $\tilde{O}(kG^2 \ln \max_t \|w_t\|)$ term. This remaining corruption bound can be controlled by adding a small multiple of the Huber regularizer $f_t$, which satisfies $\sum_t f_t(w) \geq \tilde{O}(kG^2 \ln \max_t \|w_t\|)$ even for $c \ll G$.

To summarize, our solution to solve Challenge 2 is to set regularizer as:

$$r_t(w) = f_t(w) + a_t\|w\|^2$$

where $a_t = \alpha_t + \beta_t$. Overall, the sparsely applied quadratic regulaization and small Huber regularization at every round allow the following bounds as shown in Lemma H.2:

ERROR−CORRECTION $\lesssim$

$$(kG)^2 \left(\sum_{t:\alpha_t>0} \frac{1}{\alpha_t} + \sum_{t:\beta_t>0} \frac{1}{\beta_t}\right) - \sum_{t=1}^T f_t(w_t)$$

and

$$\text{BIAS} \lesssim \|u\|^2 \left(\sum_{t=1}^T \alpha_t + \sum_{t=1}^T \beta_t\right) + \sum_{t=1}^T f_t(u) + kG\|u\|$$

with appropriately chosen constants, ERROR − CORRECTION $\leq \tilde{O}(kG^2)$ and BIAS $\leq \tilde{O}(kG\|u\| + \|u\|^2)$ are achieved.

## 6.3. Base Algorithm and the Regret

With the regularizer $r_t$ chosen in the previous section, it remains to choose a learner $\mathcal{A}$ which guarantees regret:

$$R_T^{\mathcal{A}}(u) := \sum_{t=1}^{T} \langle \tilde{g}_t^c, w_t - u \rangle + f_t(w_t) - f_t(u)$$

$$+ \sum_{t=1}^{T} a_t \left( \|w_t\|^2 - \|u\|^2 \right)$$

However, the solution of Section 5.1 of learning with composite loss $\tilde{\ell}_t(w) = \langle \tilde{g}_t^c, w_t \rangle + f_t(w) + a_t \|w\|^2$ is no longer appropriate because it requires the composite term to be known in round $t$. Unfortunately, in this case the quadratic component depends on $a_t = \alpha_t + \beta_t$, which is unknown until $\tilde{g}_t^c$ is revealed. Due to the quadratic component, we employ the "epigraph-based regularization" technique recently developed by Cutkosky & Mhammedi (2024) to construct $\mathcal{A}$. Briefly, this technique projects convex optimization problems in $\mathcal{W} = \mathbb{R}^d$ to an augmented space $\mathbb{R}^{d+1}$. The first $d$ coordinate of the solution in the augmented space is the decision variable $w_t$, and the extra coordinate is a technical device that is used to modify the loss in a way that makes it easier to control the quadratic penalty. We incorporate this technique in our set up to obtain $R_T^{\mathcal{A}}(u) \leq \tilde{O}\left( \|u\| G\sqrt{T} + \|u\|^2 k \right)$ as provided in Theorem I.3. We leave a brief summary of the technique in Appendix I.

Combining with the error correction parts from the previous section, we obtain a general regret guarantee as follows with free parameters $c, \gamma_\alpha, \gamma_\beta$ that specify the Huber regularizer and quadratic regularizers:

**Theorem 6.1.** *Suppose $g_t, \tilde{g}_t$ satisfies assumptions in Equation (3) and (4). Setting $r_t(w) = f_t(w) + \alpha_t \|w\|^2$, where $f_t$ is defined in Equation (7) with parameters: $\alpha = \epsilon \tau_G / c$, $p = \ln T$, $\gamma = \gamma_\alpha + \gamma_\beta$, for some $\epsilon, c, \gamma_\alpha, \gamma_\beta, \tau_G, \tau_D > 0$, set Algorithm 5 algorithm as the base algorithm $\mathcal{A}$. Then Algorithm 1 guarantees:*

$$R_T(u) \leq \tilde{O}\Bigg[ \epsilon[G]_{\tau_G} + k\tau_D[G]_{\tau_G} + \|u\|[G]_{\tau_G}\left(\sqrt{T} + k\right)$$

$$+ c\left(\|u\| + \tau_D\right) + \left(\gamma_\alpha(k+1) + \gamma_\beta\right)\|u\|^2$$

$$+ \frac{(k+1)[G]_{\tau_G}^2}{\gamma_\alpha} + \frac{k^2[G]_{\tau_G}^2}{\gamma_\beta}\Bigg]$$

Notice that although it appears possible to set $c = 0$ in the above bound, there is a $\log(1/c)$ term hidden inside the $\tilde{O}$ that prevents excessively small $c$ values.

Next, we provide two different way to set $c, \gamma_\beta$. By setting both $c, \gamma_\beta = O(k)$ and $\tau_D = O(1/k)$, we obtain:

**Corollary 6.2.** *With $c = k\tau_G, \gamma_\beta = k, \gamma_\alpha = 1, \tau_D = \epsilon/k$ and rest of parameters same as Theorem 6.1, Algorithm 1 guarantees a regret bound:*

$$R_T(u) \leq \tilde{O}\Bigg[ \epsilon[G]_{\tau_G} + \|u\|[G]_{\tau_G}\left(\sqrt{T} + k\right)$$

$$+ (k+1)\left(\|u\|^2 + [G]_{\tau_G}^2\right)\Bigg]$$

*where $[G]_{\tau_G} := \max(\tau_G, G)$.*

Just as in the known-$G$ case, the parameter settings in Corollary 6.2 yield $\tilde{O}(\sqrt{T})$ regret so long as $k \leq \sqrt{T}$ so that we can experience a significant amount of corruption without damaging the asymptotics of the regret bound.

We can also achieve the desirable "safety" property of Theorem 6.1 in which the regret with respect to the baseline point $u = 0$ is constant no matter what $k$ is via a different setting of the regularization parameters as provided in Corollary 6.3:

**Corollary 6.3.** *With $c = \tau_G, \gamma_\beta = k^2, \gamma_\alpha = k + 1, \tau_D = 1$ and rest of parameters same as Theorem 6.1, Algorithm 1 guarantees a regret bound:*

$$R_T(u) \leq \tilde{O}\Bigg[ \epsilon[G]_{\tau_G} + (k+1)[G]_{\tau_G} + [G]_{\tau_G}^2$$

$$+ \|u\|[G]_{\tau_G}\left(\sqrt{T} + k + 1\right) + \|u\|^2(k+1)^2\Bigg]$$

*where $[G]_{\tau_G} := \max(\tau_G, G)$.*

However, in this case we now pay a larger penalty for $u \neq 0$ that scales with $k^2$ rather than $k$. This is a natural tradeoff: we ensured constant regret at the origin by increasing the regularization away from the origin. The overall regularization is stronger, which encourages smaller $\|w_t\|$. Thus, this algorithm configuration is likely to be advantageous when competing with small $\|u\|$.

## 7. Conclusion

In this paper, we considered unconstrained online convex optimization that only have access to potentially corrupted gradients $\tilde{g}_t$ instead of the true gradient $g_t$, in which the corruption level is measured by $k$. In the case that $G \geq \max_t \|g_t\|$ is known, we provide an algorithm that achieves the optimal regret guarantee $\|u\| G(\sqrt{T} + k)$. When $G$ is unknown it incur an extra additive penalty of $(\|u\|^2 + G^2)k$. While the $\|u\|^2 + G^2$ is optimal without corruption (Cutkosky & Mhammedi, 2024), it is unclear whether the multiplicative dependence on $k$ is optimal in the presence of corruption.

## Impact Statement

This paper advances the theoretical understanding of online convex optimization, contributing to the mathematical foundations of machine learning. As it does not involve deployable systems or datasets, the broader societal and ethical implications are indirect, and no specific concerns need to be highlighted.

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

## A. Unconstrained Online Convex Optimization with Hints

In the unconstrained setting, there are algorithms requires a uniform bound $G \geq \max_t \|g_t\|$ upfront which guarantees $\tilde{O}(\|u\|G\sqrt{T})$ (McMahan & Orabona, 2014; Orabona & Pál, 2016; Cutkosky & Orabona, 2018; Zhang et al., 2022). In the case where $G$ is unknown, algorithms are usually devised through an intermediate step with a slightly ideal scenario, that is the algorithm receives a gradient $g_t$ with a "hints" $h_{t+1} = \max_{i \leq t+1} \|g_i\|$ at each iteration $t$. It turns out by having access to $h_t$ to guide the algorithm, same regret $\tilde{O}(\|u\|h_T\sqrt{T})$ can be achieved (Cutkosky, 2019; Mhammedi & Koolen, 2020; Jacobsen & Cutkosky, 2022; Zhang et al., 2024).

In this paper, we also follows the same strategy of assuming a good hints $h_t = \max_{i \leq t} \|g_t\|$ is supplied to the algorithm, and eventually investigate the scenario of only the current best estimate $h_t \approx \max_{i \leq t-1} \|g_t\|$ is available. Hence most of the proofs in the appendix are displayed in the way of relying on a time varying "hints": $0 < h_1 \leq \cdots h_T \leq h_{T+1}$ to accommodate the design of both known $G$ and unknown $G$ case.

## B. Bounds on Regularizer in Equation (7)

Our results are based on appropriate algebraic property of $f_t$ as displayed in Equation (7) and was firstly studied by Zhang & Cutkosky (2022). We re-stated relevant bounds as Lemma B.1 for completeness.

**Lemma B.1** (Lemma 11 and Lemma 13 of Zhang & Cutkosky (2022)). *Let $f_t : \mathcal{W} \to \mathbb{R}^+$ be defined as follows for some $c \geq 0, \alpha > 0$ and $p \geq 1$,*

$$f_t(w; c, p, \alpha) = \begin{cases} c(p\|w\| - (p-1)\|w_t\|) \frac{\|w_t\|^{p-1}}{(\sum_{i=1}^t \|w_i\|^p + \alpha^p)^{1-1/p}}, & \|w\| > \|w_t\| \\ c\|w\|^p \frac{1}{(\sum_{i=1}^t \|w_i\|^p + \alpha^p)^{1-1/p}}, & \|w\| \leq \|w_t\| \end{cases}$$

*Then*

$$\sum_{t=1}^T f_t(w_t) \geq c\left(\left(\sum_{t=1}^T \|w_t\|^p + \alpha^p\right)^{1/p} - \alpha\right)$$

$$\sum_{t=1}^T f_t(u) \leq cp\|u\|T^{1/p}\left[\ln\left(1 + \left(\frac{\|u\|}{\alpha}\right)^p\right)^{(p-1)/p} + 1\right]$$

*In particular, when $p = \ln T$ for $T \geq 3$:*

$$\sum_{t=1}^T f_t(w_t) \geq c\left(\max_t \|w_t\| - \alpha\right)$$

$$\sum_{t=1}^T f_t(u) \leq 3c\ln T\|u\|\left[\ln\left(1 + \left(\frac{\|u\|}{\alpha}\right)^p\right) + 2\right]$$

*Proof.* The first set of bounds are the same as Zhang & Cutkosky (2022) Lemma 13. For the second set of bounds: the lower bound is due to $\left(\sum_{t=1}^T \|w_t\|^p + \alpha^p\right)^{1/p} \geq \left(\sum_{t=1}^T \|w_t\|^p\right)^{1/p}$ followed by an application of of Lemma 11 in Zhang & Cutkosky (2022); the upper bound is due to $x^q \leq x + 1$ for $x > 0$ and $0 < q < 1$, where we set $x = \ln\left(1 + \left(\frac{\|u\|}{\alpha}\right)^p\right)$ and $q = (p-1)/p$ followed by $T^{1/\ln T} = e \leq 3$. $\square$

## C. Base Algorithms for known G

In this section, we verify the "centered mirror descent" framework (see their Algorithm 1 and we contextualize it as Algorithm 2 here) developed by (Jacobsen & Cutkosky, 2022) automatically guarantee a composite regret $R_T^{\mathcal{A}}(u)$ "for free" thus is a compatible candidate in achieving robust unconstrained learning.

Algorithm 2 is an explicit update with two regularizers $\psi_t(w)$:

$$\psi_t(w) = 3\int_0^{\|w\|} \Psi_t'(x)dx, \quad \Psi_t'(x) = \min_{\eta \leq 1/h_t}\left[\frac{\ln(x/a_t + 1)}{\eta} + \eta V_t\right] \tag{13}$$

and some $f_t(w)$ (In their notation as $\varphi_t(w)$ ). That is

$$w_{t+1} = \underset{w}{\operatorname{argmin}} \langle g_t, w \rangle + D_{\psi_t}(w \mid w_t) + \Delta_t(w) + f_t(w) \tag{14}$$

where $D_\psi(a \mid b) = \psi(a) - \psi(b) - \langle \nabla\psi(b), a - b \rangle$ is the Bregman divergence induced by $\psi$, and $\Delta_t(w) = D_{\psi_{t+1}}(w \mid w_1) - D_{\psi_t}(w \mid w_1)$.

Notice the first two terms in Equation (14) corresponds to the classical mirror descent with mirror map $\psi_t$, the third term $\Delta_t(w)$ encourages iterate to be close to the initial start $w_1$, thus echoing "centered" mirror descent. and $f_t$ being any arbitrary composite regularizer and its structure is completely known in time in order to produce $w_{t+1}$. shows the above update with $f_t = 0$ ($\varphi_t$ in their notation) guarantees $R_T(u) \leq \tilde{O}(\|u\|h_T\sqrt{T})$ (Theorem 6), which is optimal. The second composite regularizer $f_t$ was specifically tailored as $f_t(w) = O(\|g_t\|^2\|w\|)$ in order to attain optimal dynamic regret (See their Proposition 1).

We verify $R_T^A(u) \leq \tilde{O}(\|u\|h_T\sqrt{T})$ when $f_t$ as defined in Equation (7) for our purpose, which is formalized as Theorem C.2. In fact, the development of (Jacobsen & Cutkosky, 2022) attains the same bound for any arbitrary composite regularizer $f_t$ for user to exploit. We first present a helper Lemma that is equivalent to the update in Equation (14) before the theorem:

**Lemma C.1.** *Equation (14) is equivalent to*

$$\tilde{w}_{t+1} = \underset{w}{\operatorname{argmin}} \langle g_t, w \rangle + D_{\psi_t}(w \mid w_t) \tag{15}$$

$$w_{t+1} = \underset{w}{\operatorname{argmin}} D_{\psi_t}(w \mid \tilde{w}_{t+1}) + \Delta_t(w) + f_{t+1}(w) \tag{16}$$

*Proof.* By first order optimality, Equation (15) implies

$$g_t + \nabla\psi_t(\tilde{w}_{t+1}) - \nabla\psi_t(w_t) = 0$$

Thus

$$\tilde{w}_{t+1} = \nabla\psi_t^{-1}\left(\nabla\psi_t(w_t) - g_t\right)$$

Substitute $\tilde{w}_{t+1}$ into Equation (16), we see the equivalence. $\qquad\square$

**Theorem C.2.** *Suppose $0 < h_1 \leq h_2 \leq \cdots \leq h_T$ and $g_1, \cdots, g_T$ be arbitrary sequence satisfies $\|g_t\| \leq h_t$ for all $t \in [T]$. Then Algorithm 2 guarantees*

$$R_T^A(u) := \sum_{t=1}^T \langle g_t, w_t - u \rangle + r_t(w_t) - r_t(u) \leq \tilde{O}\left(\epsilon h_T + \|u\|h_T\sqrt{T}\right)$$

*where $r_t$ is defined as Equation (7) for arbitrary $c, p, \alpha > 0$.*

*Proof.* First, we verify Algorithm 2 is indeed the update corresponding to Equation (14). By Lemma C.1, Equation (14) is equivalent to a two step update as shown in Equation (15) and (16).

By first order optimality to Equation (15):

$$\tilde{w}_{t+1} = \nabla\psi_t^{-1}\left(\nabla\psi_t(w_t) - g_t\right)$$

Then substitute to Equation (16)

$$\nabla\psi_{t+1}(w_{t+1}) + \nabla f_{t+1}(w_{t+1}) = \nabla\psi_t(w_t) - g_t := \theta_t$$

Define $\Psi_t(\|w\|) := \psi_t(w) = \int_0^{\|w\|} \Psi_t'(x)dx$ and $R_{t+1}(x) := cp\frac{|x|^{p-1}}{(S_t+|x|^{p-1})^{1-1/p}}$ (Notice $\nabla f_{t+1}(x) = \frac{x}{\|x\|}R_{t+1}(\|x\|)$), thus by substitute derivatives

$$\frac{w_{t+1}}{\|w_{t+1}\|} \left( \underbrace{\Psi_{t+1}'(\|w_{t+1}\|) + R_{t+1}(\|w_{t+1}\|)}_{:=\mathcal{L}_{t+1}(\|w_{t+1}\|)} \right) = \theta_t$$

Thus, the solution $w_{t+1}$ satisfies the following for some $x \geq 0$:

$$w_{t+1} = x\frac{\theta_t}{\|\theta_t\|}, \quad \mathcal{L}_{t+1}(x) = \|\theta_t\|$$

In particular, define $F_t = \log(1 + \frac{x}{a_t})$

$$\mathcal{L}_t(x) = \begin{cases} 6\sqrt{V_t F_t(x)} + R_t(x), & h_t\sqrt{F_t(x)} < \sqrt{V_t} \\ 3h_t F_t(x) + \frac{3V_t}{h_t} + R_t(x), & \text{otherwise} \end{cases}$$

The boundary case is when $x^* : h_t\sqrt{F_t(x^*)} = \sqrt{V_t}$. That is $x^* = a_t(\exp(V_t/h_t^2) - 1)$. Consider by cases, if

$$\mathcal{L}_t(x) = \|\theta_t\| \leq \mathcal{L}_t(x^*) = \frac{6V_t}{h_t} + R_t(x^*)$$

Then $x \leq x^*, h_t\sqrt{F_t(x)} < \sqrt{V_t}$ thus the first branch should be evoked. Similarly, consider the other case, the complete update should be:

$$\mathcal{L}_t(x) = \begin{cases} 6\sqrt{V_t F_t(x)} + R_t(x), & \|\theta_t\| \leq \frac{6V_t}{h_t} + R_t(x^*) \\ 3h_t F_t(x) + \frac{3V_t}{h_t} + R_t(x), & \text{otherwise} \end{cases}$$

It remains to show the composite regret $R_T^{\mathcal{A}}(u)$ guaranteed by Algorithm 2. Notice the original proof of Theorem 6 of (Jacobsen & Cutkosky, 2022) relies on their generalized Lemma 1 by setting $f_t = 0$ (their $\varphi_t$). Thus Lemma 1 of (Jacobsen & Cutkosky, 2022) implies for general $f_t$:

$$R_T(u) \leq \psi_{T+1}(u) + \sum_{t=1}^{T} f_t(u) + \sum_{t=1}^{T} \langle g_t, w_t - w_{t+1}\rangle - D_{\psi_t}(w_{t+1} \mid w_t) - \Delta_t(w_{t+1}) - f_{t+1}(w_{t+1})$$

Move relative terms to left hand side:

$$R_T(u) + \sum_{t=1}^{T} f_t(w_t) - f_t(u) \leq \psi_{T+1}(u) + \sum_{t=1}^{T} \langle g_t, w_t - w_{t+1}\rangle - D_{\psi_t}(w_{t+1} \mid w_t) - \Delta_t(w_{t+1}) + f_1(w_1) - f_{T+1}(w_{T+1})$$

$w_1 = 0$, thus $f_1(w_1) = 0$. And $f_t$ is non-negative

$$\leq \psi_{T+1}(u) + \sum_{t=1}^{T} \langle g_t, w_t - w_{t+1}\rangle - D_{\psi_t}(w_{t+1} \mid w_t) - \Delta_t(w_{t+1})$$

Thus following the exactly same proof of Theorem 6 of (Jacobsen & Cutkosky, 2022):

$$R_T^{\mathcal{A}}(u) \leq 4h_T + 6\|u\| \max\left[\sqrt{V_{T+1}\ln\left(1 + \frac{\|u\|\sqrt{B_{T+1}\ln^2(B_{T+1})}}{\epsilon}\right)}, h_T\ln\left(1 + \frac{\|u\|\sqrt{B_{T+1}\ln^2(B_{T+1})}}{\epsilon}\right)\right]$$

$\square$

---

**Algorithm 2** Robust Online Learning By Exploiting Linear Offset

---

1: **Input:** Time horizon $T$; Initial Value $\epsilon$; Corruption parameter $k$; Regularization relevant parameters: $c, p, \alpha$;
   Hints: $0 < h_1 \le h_2 \le \cdots, \le h_{T+1}$;
2: **Initialize:** $\theta_1 = 0, C_1 = 0, N_1 = 4, B_1 = 4N_1, w_1 = 0, S_0 = \alpha^p$.
3: **for** $t = 1$ **to** $T$ **do**
4:     **Define:** $F_t(x) = \ln(1 + x/a_t), R_{t+1}(x) = cp\|x\|^{p-1}/(S_t + \|x\|^p)^{1-1/p}$
5:     Output $w_t$, receive $g_t$ where $\|g_t\| \le h_t$
6:     Compute $\theta_{t+1} = \nabla \psi_t(w_t) - g_t$
7:     Update $C_{t+1} = C_t + \|g_t\|^2, N_{t+1} = N_t + \|g_t\|^2/h_t^2, B_{t+1} = B_t + 4N_t$
8:     Set $V_{t+1} = h_{t+1}^2 + C_{t+1}$ and $\alpha_{t+1} = \frac{\epsilon}{\sqrt{B_{t+1}}\ln^2(B_{t+1})}$
9:     Compute $x^* = a_{t+1}\left(\exp(V_{t+1}/h_{t+1}^2) - 1\right)$
10:    Define

$$
\mathcal{L}_{t+1}(x) = \begin{cases} 6\sqrt{V_{t+1}F_{t+1}(x)} + R_{t+1}(x), & \|\theta_t\| \le \frac{6V_{t+1}}{h_{t+1}} + R_{t+1}(x^*) \\ 3h_{t+1}F_{t+1}(x) + \frac{3V_{t+1}}{h_{t+1}} + R_{t+1}(x), & \text{otherwise} \end{cases}
$$

11:    Solve for $x_{t+1} : \mathcal{L}_{t+1}(x_{t+1}) = \|\theta_t\|$             {solvable via bisection where $x_{t+1} \in [0, S_t^{1/p}]$}
12:    Update $w_{t+1} = x_{t+1}\frac{\theta_t}{\|\theta_t\|}, S_{t+1} = S_t + \|w_{t+1}\|^p$
13: **end for**

---

We remark that the inverse problem involved in Algorithm 2 Line 11 can be efficiently handled by bisection method since $\mathcal{L}_{t+1}(x)$ is monotonically increasing. Notice that $0 = \mathcal{L}_{t+1}(0) \le \mathcal{L}_{t+1}(x_{t+1}) \le 0$, so 0 can be used as an initial lower bound. On the other hand $R_{t+1}(x_{t+1}) \le \mathcal{L}_{t+1}(x_{t+1}) = \|\theta_t\|$, thus $x_{t+1} \le R_{t+1}^{-1}(\|\theta\|)$, where for $y \ge 0$:

$$
R_{t+1}^{-1}(y) = \left( \frac{S_t\,(y/cp)^{\frac{p}{p-1}}}{1 + (y/cp)^{\frac{p}{p-1}}} \right)^{\frac{1}{p}}
$$

For simplicity, we can always use $S_t^{1/p}$ as an initial upper bound.

## D. Proof to Theorem 5.1 and Applications

We provide the regret guarantee of Algorithm 1 when using an instance of Algorithm 2 as a base learner $\mathcal{A}$ as well as two applications when $h_1 = \cdots = h_{T+1} = G$

**Theorem 5.1.** *Suppose $g_t, \tilde{g}_t$ satisfies assumptions in Equation (3) and (4). Setting $h_t = G, r_t = f_t$ as defined in Equation (7) with $c = kG, p = \ln T, \alpha = \epsilon/k$ for some $\epsilon > 0$, set Algorithm 2 algorithm as the base algorithm $\mathcal{A}$. Then Algorithm 1 guarantees:*

$$
R_T(u) \le \tilde{O}\left[\epsilon G + \|u\|G\left(\sqrt{T} + k\right)\right]
$$

*Proof.* The proof begins with the regret decomposition in Equation (6) and is displayed below for convenience.

$$
R_T(u) := \underbrace{\sum_{t=1}^T \langle \tilde{g}_t^c, w_t - u \rangle + r_t(w_t) - r_t(u)}_{:=R_T^{\mathcal{A}}(u)} + \underbrace{\sum_{t=1}^T \langle g_t - \tilde{g}_t^c, w_t \rangle - \sum_{t=1}^T r_t(w_t)}_{:=\text{ERROR}} \underbrace{\vphantom{\sum_{t=1}^T}}_{:=\text{CORRECTION}} + \underbrace{\left\langle \sum_{t=1}^T g_t - \tilde{g}_t^c, -u \right\rangle + \sum_{t=1}^T r_t(u)}_{:=\text{BIAS}}
$$

Define OFFSET $:=$ ERROR $-$ CORRECTION, and in Section 5.1 we already shown ERROR $\le kG\max_t \|w_t\|$, thus

$$
\text{OFFSET} \le kG\max_t \|w_t\| - \sum_{t=1}^T r_t(w_t)
$$

by Lemma B.1 by substituting $c, p\alpha$

$$\leq kG \max_t \|w_t\| - kG(\max_t |w_t| - \epsilon/k) = O(\epsilon G)$$

Similarly by following the application of Lemma B.1:

$$\text{BIAS} \leq kG\|u\| + \sum_{t=1}^{T} r_t(u)$$

$$\leq kG\|u\| + 3kG \ln T |u| \left[ \ln \left( 1 + \left( \frac{|u|k}{\epsilon} \right)^{\ln T} \right) + 2 \right]$$

$$= \tilde{O}(kG\|u\|)$$

In addition, Algorithm 2 is set as $\mathcal{A}$ in response to $\tilde{g}_t^c$ with $h_t = G$. Thus by Theorem C.2 guarantees

$$R_T^{\mathcal{A}}(u) \leq \tilde{O}(\epsilon G + \|u\|G\sqrt{T})$$

Combining three parts and we complete the proof. $\square$

Here, we provide implication of Theorem 5.1 to stochastic convex optimization and an online version of distributionally robust optimization.

**Stochastic convex optimization with corruptions**  OCO and convex stochastic optimization are connected through the classical Online-to-Batch Conversion (Orabona, 2019). Below, we present the implications of Theorem 5.1 stochastic convex optimization in a setting where $k$ gradient evaluations are arbitrarily corrupted.

**Corollary D.1** (Stochastic Convex Optimization via Online to Batch). *Suppose $\mathcal{L} : \mathcal{W} \rightarrow \mathbb{R}$ is convex and $\mathbb{E}[\ell_t(w)] = \mathcal{L}(w), g_t = \nabla\ell_t(w_t)$ and $\mathbb{E}_t[\|g_t\|] \leq G$. Algorithm 1 have access to $\tilde{g}_t$ such that $\sum_{t=1}^{T} \mathbb{1}\{g_t \neq \tilde{g}_t\} \leq k$, then Algorithm 1 guarantees*

$$\mathbb{E}\left[ \mathcal{L}\left( \frac{\sum_{t=1}^{T} w_t}{T} \right) - \mathcal{L}(u) \right] \leq \tilde{O}\left[ \frac{\epsilon + \|u\|G\left(\sqrt{T} + k\right)}{T} \right]$$

*Proof.* The proof leverages the standard online to batch conversion (Theorem 3.1 in (Orabona, 2019) by setting $\alpha_t = 1$), then combining with the regret bounds from Theorem 5.1. $\square$

**Distributionally robust optimization**  Distributionally robust optimization is a form of robust stochastic optimization on training data sampled from distribution $P$ that is not the same as the population distribution $Q$ (Ben-Tal et al., 2009; 2015). Typically, $Q$ is considered as uniform, but the actual training data collection process might be biased, meaning $P$ is different to $Q$. In this situation, stochastic optimization which treats each training example with equal weight is no longer appropriate.

Namkoong & Duchi (2016) formalized this framework as the following model with respect to a set of losses $\ell_1, \ldots \ell_T$, and an uncertainty set $\mathcal{P}_k = \{P \in \Delta^T : D_f(P\|Q) \leq C(k, T)\}$, where $D_f(P\|Q)$ is the $f$-Divergence, for a convex function $f : \mathbb{R}^+ \mapsto \mathbb{R}$ with $f(1) = 0$.

$$\underset{w}{\text{argmin}} \sup_{P \in \mathcal{P}_k} \sum_{t=1}^{T} p_t \ell_t(w)$$

the decision variable from above formulation takes account into the worst case distributional uncertainty, hence is intuitively associated with improving generalization error given an appropriate uncertainty set $\mathcal{P}_k$ (Sagawa et al., 2019).

Distributionally robust optimization is increasingly relevant in the training of large language models, where training data are sourced from different domains (Xie et al., 2023). This is due to data from some domain are relatively atypical in comparison to others in representing the overall population distribution (Oren et al., 2019). Although empirical gain has been observed by incorporating distributionally robust optimization, the scalability has always been a primary concern for

model training (Levy et al., 2020; Qi et al., 2021). Therefore, we consider a natural "online" version of distributionally robust optimization model proposed by Namkoong & Duchi (2016), with its online analogous metric formulated as:

$$\sup_{P \in \mathcal{P}_k} \sum_{t=1}^{T} p_t(\ell_t(w_t) - \ell_t(u))$$

We present the implication of Algorithm 1 to this problem with respect to total variation $D_{TV}$ and Kullback-Leibler divergence $D_{KL}$. In particular, we assume $\ell_t$ is convex and $Q$ is uniform.

**Corollary D.2** (Online Distributionally Robust Optimization). *Suppose $\tilde{g}_t \in \nabla \ell_t(w_t)$ and $\|\tilde{g}_t\| \leq G$. Algorithm 1 runs on $\tilde{g}_t$ guarantees*

$$\sup_{P \in \mathcal{P}_k} \sum_{t=1}^{T} p_t(\ell_t(w_t) - \ell_t(u)) \leq \tilde{O}\left[\frac{\epsilon + \|u\|G\left(\sqrt{T} + k\right)}{T}\right]$$

*for $D_{TV} \leq \frac{k}{T}$. In addition, in the case where $D_{KL} \leq \frac{2k^2}{T^2}$ the same guarantee is achieved.*

*Proof.* We begin with the case of $D_{TV}(P\|Q) = \frac{1}{2}\sum_{t=1}^{T} q_t |\frac{p_t}{q_t} - 1| \leq \frac{k}{T}$, where $q_t = \frac{1}{T}$. First, we link the regret incurred by Algorithm 2 that runs on $g_t$, and we denote the *unobservable* gradient as $\tilde{g}_t = \frac{p_t}{q_t} g_t$

$$\begin{aligned}
\tilde{R}_T(u) := \sum_{t=1}^{T} p_t(\ell_t(w_t) - \ell(u)) &\leq \sum_{t=1}^{T} p_t \langle g_t, w_t - u \rangle \\
&= \sum_{t=1}^{T} q_t \langle g_t, w_t - u \rangle + \sum_{t=1}^{T} q_t \left(\frac{p_t}{q_t} - 1\right) \langle g_t, w_t - u \rangle \\
&= \frac{1}{T}\left(\sum_{t=1}^{T} \langle g_t, w_t - u \rangle + \sum_{t=1}^{T} \langle \tilde{g}_t - g_t, w_t - u \rangle\right)
\end{aligned}$$

since $\frac{1}{G}\sum_{t=1}^{T} \|g_t - \tilde{g}_t\| \leq \sum_{t=1}^{T} |1 - \frac{p_t}{q_t}| \leq 2k$, $\tilde{g}_t, g_t$ satisfies Equation (2), hence Theorem 5.1 provides the guarantee:

$$\sum_{t=1}^{T} p_t(\ell_t(w_t) - \ell(u)) \leq \tilde{O}\left[\frac{\epsilon + \|u\|G\left(\sqrt{T} + k\right)}{T}\right]$$

In terms of $D_{KL}$, we exploit the Pinsker's inequality $D_{TV} \leq \sqrt{2D_{KL}}$, Hence $D_{KL} \leq \frac{2k^2}{T^2}$ yields to the same results. $\square$

# E. Lower Bounds

In this section, we present two type of matching lower bounds to Theorem 5.1: Theorem 5.2 provides a lower bound for any comparator $u^* \in \mathbb{R}^d$ with arbitrary magnitude $D > 0$. Theorem E.5 is a lower bound with log factors, which appears in unconstrained OCO upper bounds.

We begin by presenting a helper lemma that aids in the analysis of Theorem 5.2, followed by Lemmas required to proof to Theorem 5.2.

**Lemma E.1.** *Suppose $z_1, z_2, \cdots, z_T \in \{-1, +1\}$ with equal probability. Then for every $t \in [T]$ for some $T \geq 1$.*

$$\mathbb{E}\left[\sum_{t=1}^{T} sign\left(\sum_{i=1}^{T} z_i\right) z_t\right] \geq \sqrt{\frac{T}{16}}$$

*Proof.* Define $S_t = \sum_{i \in [T]:i \neq t} z_i$, by conditioning on $g_T \in \{-1, +1\}$:

$$2 \mathbb{E} \left[ \text{sign} \left( \sum_{i=1}^{T} z_T \right) z_t \right] = \mathbb{E} \left[ \text{sign} \left( S_T + 1 \right) \right] - \left[ \text{sign} \left( S_T - 1 \right) \right]$$

$$= \sum_{k \in \{-T, -T+2, \cdots, T\}} (\text{sign}(k+1) - \text{sign}(k-1)) P(S_T = k)$$

We consider $T$ by cases: suppose $T$ is even, $\text{sign}(k+1) - \text{sign}(k-1) = 2$ when $k = 0$, and $\text{sign}(k+1) - \text{sign}(k-1) = 0$ otherwise. Thus applying $\binom{T}{T/2} \geq 2^{T-1} (T/2)^{-1/2}$

$$\mathbb{E} \left[ \text{sign} \left( \sum_{i=1}^{T} z_i \right) z_T \right] = P(S_T = 0) = \binom{T}{T/2} 2^{-T} \geq 2^{-1} (T/2)^{-1/2} = \sqrt{\frac{1}{2T}}$$

Similarly if $T$ is odd, by symmetry to $S_T = \pm 1$:

$$\mathbb{E} \left[ \text{sign} \left( \sum_{i=1}^{T} z_i \right) z_T \right] = \frac{1}{2} \left( P(S_T = -1) + P(S_T = 1) \right)$$

$$= \binom{T}{(T+1)/2} 2^{-T}$$

Define $T' = T - 1$ thus $T'$ is even

$$= \frac{T'!}{\left( \frac{T'}{2} \right)! \left( \frac{T'}{2} \right)!} \cdot \frac{\left( \frac{T'}{2} \right)! \left( \frac{T'}{2} \right)!}{\left( \frac{T'+2}{2} \right)! \left( \frac{T'}{2} \right)!} \cdot \frac{(T'+1)!}{(T')!} 2^{-(T'+1)}$$

$$= \binom{T'}{T'/2} \frac{T'+1}{\frac{T'+2}{2} + \frac{T'}{2}} 2^{-(T'+1)}$$

$$\geq 2^{T'-1} \left( \frac{T'}{2} \right)^{-1/2} \frac{T'+1}{T'+2} 2^{-(T'+1)}$$

$$\geq \frac{1}{8T'} = \frac{1}{8(T-1)} \geq \frac{1}{16T}$$

Thus combining two cases:

$$\mathbb{E} \left[ \text{sign} \left( \sum_{i=1}^{T} z_i \right) z_T \right] \geq \frac{1}{16T}$$

Due to symmetry, $S_t$ has the same distribution $\forall t \in [T]$:

$$\mathbb{E} \left[ \text{sign} \left( \sum_{i=1}^{T} z_T \right) z_t \right] = \mathbb{E} \left[ \text{sign} \left( \sum_{i=1}^{T} z_i \right) z_T \right], \quad \forall t \in [T]$$

Thus

$$\mathbb{E} \left[ \sum_{t=1}^{T} \text{sign} \left( \sum_{i=1}^{T} z_i \right) z_t \right] = T \mathbb{E} \left[ \text{sign} \left( \sum_{i=1}^{T} z_i \right) z_T \right] \geq \sqrt{\frac{T}{16}}$$

$\square$

**Theorem 5.2.** *For every $D > 0$, there exists a comparator $u^* \in \mathbb{R}^d$ such that $\|u^*\| = D$, $\tilde{g}_1, \cdots, \tilde{g}_T$ and $g_1, \cdots, g_T$ such that $\|g_t\|, \|\tilde{g}_t\| \leq 1$, $\sum_{t=1}^{T} \mathbb{1}\{\tilde{g}_t \neq g_t\} = k$:*

$$\sum_{t=1}^{T} \langle g_t, w_t - u^* \rangle \geq \Omega \left[ \|u^*\| \left( \sqrt{T} + k \right) \right]$$

*Proof.* Consider the following random sequence: $z_{k+1}, z_{k+2}, \cdots, z_T \in \{-1, +1\}$ with equal probability and $z_1 = \cdots, z_k = \text{sign}(\sum_{t=k+1}^T z_t)$. And $\tilde{z}_1 = \cdots = \tilde{z}_k = 0$ and $\tilde{z}_t = z_t, \forall t \geq k+1$. Let $q \in \mathbb{R}^n$ be any unity vector. Suppose $g_t = z_t q, \tilde{g}_t = \tilde{z}_t q, \forall t \in T$. Select $u^* = -D \text{sign}(\sum_{t=k+1}^T g_t) q$. Thus:

$$
\begin{aligned}
\mathbb{E}[R_T(u^*)] &= \mathbb{E}\left[\sum_{t=1}^T \langle g_t, w_t - u\rangle\right] \\
&= \sum_{t=1}^T \mathbb{E}\left[\langle g_t, w_t\rangle\right] - \mathbb{E}\left[\sum_{t=1}^k \langle g_t, u\rangle\right] - \sum_{t=k+1}^T \mathbb{E}\left[\langle g_t, u\rangle\right] \\
&= \sum_{t=1}^T \mathbb{E}\left[\langle \mathbb{E}_t[z_t]q, w_t\rangle\right] + Dk + D\sum_{t=k+1}^T \mathbb{E}\left[z_t \text{sign}\left(\sum_{t=k+1}^T z_t\right)\right] \\
&= Dk + D\sum_{t=k+1}^T \mathbb{E}\left[z_t \text{sign}\left(\sum_{t=k+1}^T z_t\right)\right]
\end{aligned}
$$

by Lemma E.1

$$
\geq D\left(k + \sqrt{\frac{T-k}{16}}\right) = \Omega(\|u^*\|(k + \sqrt{T}))
$$

$\square$

The second lower bound in Theorem E.5 has a matching log factors by uses the definition of "regret at the origin" of an online learning algorithm, formalized as:

$$
R_T(0) = \sum_{t=1}^T \langle g_t, w_t - 0\rangle \leq \epsilon \tag{17}
$$

This condition implies that an algorithm maintaining small $\epsilon$ is inherently conservative: it will perform well if the comparator is close to the origin, but this behavior may come at the cost of performing poorly if the comparator is far from the origin. Before presenting the analysis to Theorem E.5, we first list previously established result on properties of iterates $w_t$ produced by any algorithm has constant regret guarantee at the origin as defined in Equation (17). Lemma E.2 was originally appeared in (Cutkosky, 2018) then being re-interpreted by (Orabona, 2019). Lemma E.3 from (Zhang & Cutkosky, 2022).

**Lemma E.2** (Theorem 5.11 of (Orabona, 2019)). *For any OLO algorithm suffers constant regret at the origin (Equation (17)) and $|g_t| \leq 1$, there exist $\beta_t \in R^d$ such that $\|\beta_t\| \leq 1$ and*

$$
w_t = \beta_t \left(\epsilon - \sum_{i=1}^{t-1} \langle g_i, w_i\rangle\right)
$$

*for all $t \in [T]$.*

**Lemma E.3** (Lemma 8 of (Zhang & Cutkosky, 2022): Unconstrained OLO Iterate Growth). *Suppose assumptions in Lemma E.2 is satisfied. Then for every $t \in [T]$, $\|w_t\| \leq \epsilon 2^{t-1}$.*

We first derive an lower bound for algorithms satisfies assumption in Lemma E.2. The construction was originally appeared in Theorem 5.12 from (Orabona, 2019). Finally, the lower bound in the context of adversarial corruptions is presented in Theorem 5.2.

**Lemma E.4** (Unconstrained OLO Lower Bound). *Suppose assumptions in Lemma E.2 is satisfied, then set $g_t = [g_{t,1}, 0, \cdots, 0]$, $g_{t,1} = g = 1$ for all $t \in [T]$. Then there exists an $u^* \in \mathbb{R}^d$ such that $\|u^*\| = 2\epsilon e^T$, and*

$$
\sum_{t=1}^T \langle g_t, w_t - u^*\rangle \geq \epsilon + \|u^*\|\sqrt{\frac{T}{30} \ln\left(1 + \frac{\|u^*\|^2 T}{2\epsilon^2}\right)}
$$

*Proof.* Let $r_t = -\sum_{i=1}^{t} \langle g_i, w_i \rangle$. Then

$$\epsilon - \sum_{t=1}^{T} \langle g_t, w_t \rangle = \epsilon + r_{T-1} - \langle g_T, w_T \rangle$$

by Lemma E.2, there exists some $\beta_T : \|\beta_T\| \leq 1$

$$= \epsilon + r_{T-1} - \langle g_T, \beta_T \rangle (\epsilon + r_{T-1})$$
$$= (1 - \langle g_t, \beta_t \rangle) (\epsilon + r_{T-1})$$

Then recursively expand $r_{T-1}, r_{T-2}, \cdots, r_1$ with Lemma E.2, then for some $\beta_t : \|\beta_t\| \leq 1$

$$\epsilon - \sum_{t=1}^{T} \langle g_t, w_t \rangle = \epsilon \prod_{t=1}^{T} (1 - \langle g_t, \beta_t \rangle)$$

Hence

$$\epsilon - \sum_{t=1}^{T} \langle g_t, w_t \rangle \leq \epsilon \prod_{t=1}^{T} \max_{\|\beta_t\| \leq 1} (1 - \langle g_t, \beta_t \rangle) = \epsilon \prod_{t=1}^{T} (1 + |g|) = \epsilon \left( 1 + \frac{|g|^2 T}{T} \right)^T \leq \epsilon \exp \left( |g|^2 T \right)$$

where we used inequality $(1 + \frac{x}{n})^n \leq e^x$ by setting $n = T, x = |g|^2 T$ for the last step. Rearrange above equation, we have

$$\sum_{t=1}^{T} \langle g_t, w_t \rangle - \epsilon \geq -\epsilon \exp \left( |g|^2 T \right) = -\epsilon \exp \left( \frac{|\sum_{t=1}^{T} g_{t,1}|^2}{T} \right) = -f(-\sum_{t=1}^{T} g_{t,1})$$

where $f(x) = \epsilon \exp(\frac{x^2}{T})$, by Theorem K.2 part 1, we have $f(x) = f^{**}(x)$. Then by the definition of double conjugate $f^{**}$,

$$\sum_{t=1}^{T} \langle g_t, w_t \rangle - \epsilon \geq -f^{**}(-\sum_{t=1}^{T} g_{t,1}) = - \left( \sup_{u_1 \in \mathbb{R}} \langle -\sum_{t=1}^{T} g_{t,1}, u_1 \rangle - f^*(u_1) \right) \tag{18}$$

By Theorem K.2 part 2, the supreme is achieve at

$$u_1^* = \nabla f(-\sum_{t=1}^{T} g_{t,1}) = \frac{2\epsilon}{T} \left( \sum_{t=1}^{T} g_{t,1} \right) \exp \left( \frac{\left( \sum_{t=1}^{T} g_{t,1} \right)^2}{T} \right) = 2\epsilon e^T$$

Substitute $u_1^*$ and set $u^* = [u_1^*, 0, \cdots, 0]$, then Equation (18) becomes:

$$\sum_{t=1}^{T} \langle g_t, w_t \rangle - \epsilon \geq \sum_{t=1}^{T} \langle g_{t,1}, u_1^* \rangle + f^*(u_1^*) = \sum_{t=1}^{T} \langle g_t, u^* \rangle + f^*(u_1^*)$$

Rearrange we have

$$\sum_{t=1}^{T} \langle g_t, w_t - u^* \rangle \geq \epsilon + f^*(u_1^*) \tag{19}$$

It remains to obtain a lower bound to $f^*(u_1^*)$. By Lemma K.4 and Lemma K.3, we have

$$f^*(u_1^*) = \sqrt{\frac{T}{2}} |u_1^*| \left( \sqrt{W \left( \frac{T|u_1^*|^2}{2\epsilon^2} \right)} - \frac{1}{\sqrt{W \left( \frac{T|u_1^*|^2}{2\epsilon^2} \right)}} \right)$$

$$\geq \sqrt{\frac{T}{2}} |u_1^*| \left( \sqrt{0.6 \ln \left( 1 + \frac{T|u_1^*|^2}{2\epsilon^2} \right)} - \frac{1}{\sqrt{0.6 \ln \left( 1 + \frac{T|u_1^*|^2}{2\epsilon^2} \right)}} \right)$$

Notice that $0.6 \ln\left(1 + \frac{T|u_1^*|^2}{2\epsilon^2}\right) = 0.6 \ln(1 + 2\exp(T)^2 T) > 1.5$, hence by Lemma K.5

$$\geq \sqrt{\frac{T}{2}} |u_1^*| \sqrt{\frac{0.2}{3} \ln\left(1 + \frac{T|u_1^*|^2}{2\epsilon^2}\right)}$$

$$= |u_1^*| \sqrt{\frac{T}{30} \ln\left(1 + \frac{T|u_1^*|^2}{2\epsilon^2}\right)}$$

Substitute the lower bound to $f^*(u_1^*)$ to Equation (19)

$$\sum_{t=1}^{T} \langle g_t, w_t - u^* \rangle \geq \epsilon + |u_1^*| \sqrt{\frac{T}{30} \ln\left(1 + \frac{|u_1^*|^2 T}{2\epsilon^2}\right)} = \epsilon + \|u^*\| \sqrt{\frac{T}{30} \ln\left(1 + \frac{\|u^*\|^2 T}{2\epsilon^2}\right)}$$

$\square$

**Theorem E.5.** *For any algorithm that maintains Equation (17) for some $\epsilon > 0$, there exists a sequence of $\tilde{g}_1, \cdots, \tilde{g}_T$ and $g_1, \cdots, g_T$ such that $\|g_t\|, \|\tilde{g}_t\| \leq 1$, $\sum_{t=1}^{T} \mathbb{1}\{\tilde{g}_t \neq g_t\} = k$, and a $u^* \in \mathbb{R}^d$ such that*

$$\sum_{t=1}^{T} \langle g_t, w_t - u^* \rangle \geq \tilde{\Omega}\left[\epsilon + \|u^*\|\left(\sqrt{T} + k\right)\right]$$

*Proof.* the proof strategy is that algorithm with regret guarantee as shown in Equation (17) attains a matching lower bound $\tilde{\Omega}(\epsilon + \|u\|\sqrt{T})$ in responding to $\mathbf{g}_t$ as shown in Lemma E.4. The by reversing the direction of exactly $k$ gradients by taking account into the growth behavior of $w_t$ (Lemma E.3) and a particular hard comparator $u^*$ constructed in Lemma E.4, we can show regrets during those rounds builds up linearly. Let $\tilde{g}_1, \cdots, \tilde{g}_T$, where $\|\tilde{g}_t\| \leq 1$ as defined in Lemma E.4 and suppose algorithm operates on those gradients. Let $S$ be the index set $S = \{t \in [T] : g_t \neq \tilde{g}_t\}$. Then by the lower bound presented in Lemma E.4

$$\sum_{t=1}^{T} \langle g_t, w_t - u^* \rangle = \sum_{t=1}^{T} \langle \tilde{g}_t, w_t - u^* \rangle + \sum_{t=1}^{T} \langle g_t - \tilde{g}_t, w_t - u^* \rangle$$

$$\geq \tilde{\Omega}(\epsilon + \|u^*\|\sqrt{T}) + \sum_{t \in S} \langle g_t - \tilde{g}_t, w_t - u^* \rangle$$

for some $u^* \in \mathbb{R}^d$ and $\|u^*\| = 2\epsilon e^T$. For $t \in S$, define $g_t$ as follows

$$g_t = \tilde{g}_t - \frac{u^*}{\|u^*\|}$$

Then

$$\sum_{t=1}^{T} \langle g_t, w_t - u^* \rangle \geq \tilde{\Omega}(\epsilon + \|u^*\|\sqrt{T}) + \sum_{t \in S} \langle -\frac{u^*}{\|u^*\|}, w_t \rangle + \sum_{t \in S} \langle \frac{u^*}{\|u^*\|}, u^* \rangle$$

$$\geq \tilde{\Omega}(\epsilon + \|u^*\|\sqrt{T}) - \sum_{t \in S} \|w_t\| + k\|u^*\|$$

Finally, By Lemma E.3 $\|w_t\| \leq \epsilon 2^{t-1}$. Hence $\|w_t\| \leq \frac{1}{2}\|u^*\|$

$$\sum_{t=1}^{t} \langle g_t, w_t - u^* \rangle \geq \tilde{\Omega}(\epsilon + \|u^*\|\sqrt{T}) - \frac{k}{2}\|u^*\| + k\|u^*\| = \tilde{\Omega}\left(\epsilon + \|u^*\|\left(\sqrt{T} + k\right)\right)$$

$\square$

## F. Adaptive Thresholding

In this section, we formalize the adaptive thresholding and clipping mechanism, namely FILTER, summarized in Section 6.2. This mechanism relies on prior knowledge of big corrupted gradients numbers which is naturally restricted by corruption model in Equation (3). We present this result as Lemma F.1, followed FILTER as Algorithm 3 and its property in Lemma F.2.

**Lemma F.1.** *For $g_1, \cdots, g_T$ and $\tilde{g}_1, \cdots, \tilde{g}_T$ that satisfies Equation (3), then there are at most $k$ number of $\tilde{g}_t$ such that $\|\tilde{g}_t\| \geq 2G$.*

*Proof.* By definition of $\mathcal{B} = \{t \in [T] : \|g_t - \tilde{g}_t\| > G\}$:

$$\begin{aligned}
\mathcal{B} &:= \{t \in [T] : \|g_t - \tilde{g}_t\| > G\} \\
&= \{t \in [T] : \|g_t - \tilde{g}_t\| > G, \|g_t\| < G\} \cup \{t \in [T] : \|g_t - \tilde{g}_t\| > G, \|g_t\| = G\} \\
&\supseteq \{t \in [T] : \|g_t - \tilde{g}_t\| > G, g_t = G \cdot \text{sign}(\tilde{g}_t)\} \\
&= \{t \in [T] : \|G - \|\tilde{g}_t\|\| > G\} \\
&= \{t \in [T] : \|\tilde{g}_t\| > 2G\}
\end{aligned}$$

Finally, due to Equation (3), $k := |\mathcal{B}| \geq |\{t \in [T] : \|\tilde{g}_t\| > 2G\}|$. $\qquad\square$

---

**Algorithm 3** FILTER: $k$-lag Thresholding and Gradient Clipping

---

1: **Input:** Corruption parameter $k$, Initial Lipschitz guess: $\tau = \tau_G > 0$.
2: **Initialize:** Filter threshold $h_1 = \tau$, $\mathcal{P} = \{\}$.
3: **for** $t = 1$ **to** $T$ **do**
4:     Receive $\tilde{g}_t$.
5:     **if** $\|\tilde{g}_t\| > h_t$ **then**
6:         Set $\tilde{g}_t^c = \frac{\tilde{g}_t}{\|\tilde{g}_t\|} h_t$, update counter: $n = n + 1$.
7:         **if** $n = k$ **then**
8:             Update Threshold $h_{t+1} = 2h_t$, reset counter: $n = 0$.
9:         **end if**
10:    **else**
11:        Set $\tilde{g}_t^c = \tilde{g}_t$, register rounds $\mathcal{P} = \mathcal{P} \cup \{t\}$.
12:        Maintain threshold $h_{t+1} = h_t$.
13:    **end if**
14:    Output $\tilde{g}_t^c, h_{t+1}$.
15: **end for**

---

We display some convenience property of Algorithm FILTER, notice all quantities apart from $h_t$ are for assisting analysis only

**Lemma F.2.** *(Algorithm 3 property) Suppose $g_t, \tilde{g}_t$ satisfies Equation (3), and Algorithm 3 receives $\tilde{g}_t$, then its per iteration outputs $\tilde{g}_t^c, h_{t+1}$ satisfies:*

*(1) $h_{t+1} = h_t, \forall t \in \mathcal{P} = \{t \in [T] : \tilde{g}_t^c = \tilde{g}_t\}$*

*(2) $\|\tilde{g}_t^c\| \leq h_t, \forall t \in [T]$*

*(3) $\tau = h_1 \leq h_2 \leq \cdots \leq h_{T+1} \leq \max(\tau, 4G)$*

*(4) $|\mathcal{P}| \geq T - (k+1) \max\left(\lceil \log_2 \frac{8G}{\tau} \rceil, 1\right)$*

*Proof.* We show each property in turns.

(1) guaranteed by algorithm line 11-12.

(2) either line 4 or line 11 is evoked to compute $\tilde{g}_t^c$.

(3) $h_t$ being non-decreasing sequence and $h_t = \tau$ is by construction. Hence it remains to show an upperbound to $h_t, \forall t \in [T+1]$. The key to this proof is there are at most $k$ number of $\tilde{g}_t$ such that $\|\tilde{g}_t\| \geq 2G$ gaurateed by Equation (3) (See Lemma F.1).

In the case where initial value of $\tau \geq 2G$, then the check point $h$ never doubled since each time of doubling requires $k+1$ number of $\|\tilde{g}_t\|$ exceeds current one. (by line 4-9)

Now, we consider $\tau < 2G$, where threshold doubling $h_{t+1} = 2h_t$ was evoked at least once (line 8) with initial value $\tau$. Then $h_{T+1} = 2^N \tau$ for some $N \in [T]$, where $N$ is the number of time line 8 was evoked.

On the other hand, at least $k+1$ number of $\tilde{g}_t$ such that $\|\tilde{g}_t\| \geq 2^{N-1}\tau$ were observed thus have triggered line 8 so eventually $h_{T+1} = 2^N \tau$. Thus by Lemma F.1, $2^{N-1}\tau \leq 2G$, $N \leq \log_2 \frac{4G}{\tau}$.

Thus $h_{T+1} = 2^N \tau \leq 4G$. Moreover, $h_t$ is non-decreasing, and we complete the proof.

(4) $|\mathcal{P}|$ is associated with the number of time in which check point $h$ doubled. By the proof to property (3) that $2^{N-1}\tau \leq 2G$, thus $N \leq \max\left(\lceil\log_2 \frac{4G}{\tau}\rceil, 0\right)$ as an upper bound that the number of line 8 being executed.

Each execution of line 8 requires exactly $k+1$ number of $\tilde{g}_t$ being clipped. Thus there were $(k+1)\max\left(\lceil\log_2 \frac{4G}{\tau}\rceil, 0\right)$ number of rounds not being register to $\mathcal{P}$ by the time when last time step $t^* \in [T]$, when the execution of line 8 happens.

For $T \geq t > t^*$, there were less than $(k+1)$ number of $\tilde{g}_t$ not being registered into $\mathcal{P}$, otherwise threshold would have been doubled. Thus

$$|\mathcal{P}| \leq (k+1)\max\left(\left\lceil\log_2 \frac{4G}{\tau}\right\rceil, 0\right) + (k+1) = (k+1)\max\left(\left\lceil\log_2 \frac{8G}{\tau}\right\rceil, 1\right)$$

$\square$

## G. Adaptive Tracking

We introduce TRACKER, a simple doubling mechanism for estimating $\max_t \|w_t\|$. as shown in Algorithm 4. The properties of TRACKER is displayed in Lemma G.1.

---

**Algorithm 4** TRACKER: Track the Magnitude of $w_t$

---

1: **Input:** Initial magnitude guess: $\tau = \tau_D > 0$.
2: **Initialize:** Filter threshold $z_1 = \tau$, (Counter, Set): $(n = 0, \mathcal{T}_n = \{\})$, Checkpoint $t_0 = 1$.
3: **for** $t = 1$ **to** $T$ **do**
4:     Receive $w_t$.
5:     **if** $\|w_t\| > z_t$ **then**
6:         Double: $z_{t+1} = 2\|w_t\|$.
7:         Update counter: $n = n + 1$.
8:         Add a new checkpoint: $t_n = t$, initialize a new set: $\mathcal{T}_n = \{\}$.
9:     **else**
10:         Maintain: $z_{t+1} = z_t$.
11:     **end if**
12:     Register round: $\mathcal{T}_n \leftarrow \mathcal{T}_n \cup \{t\}$.
13: **end for**

---

**Lemma G.1.** *(Algorithm 4 property) Algorithm 4 guarantees*

*(1) $[T]$ is partitioned by $\mathcal{T}_0, \mathcal{T}_1, \mathcal{T}_2, \cdots, \mathcal{T}_N$, for some $N$ where $N \leq \max(0, \log_2 2\max_t \|w_t\|/\tau)$.*

*(2) $\tau = z_t = z_{t+1}, \|w_t\| \leq \tau, \forall t \in \mathcal{T}_0$*

*(3) $\|w_t\| \leq 2\|w_{t_n}\|, \forall t \in \mathcal{T}_n, n \in [N]$*

*(4) $\tau = z_1 \leq z_2 \leq \cdots \leq z_{T+1} \leq \max(\tau, 2\max_t \|w_t\|)$*

*Proof.* We show each property in turns.

(1) partition property can be seen by in the initialization of $n = 0$ with increment of 1 (line 6) and whenever counter $n$ updates a new set $\mathcal{T}_n$ is created (line 7). And $\forall t \in [T]$ is assigned to $\mathcal{T}_n$ for some $n \geq 0$ (line 11).

As $z_{T+1} = 2^N \tau \leq 2 \max_t \|w_t\|$. Thus $N \leq \max(0, \log_2 2 \max_t \|w_t\|/\tau)$.

(2) For the time period of $n = 0$, line 4 was never executed.

(3) By construction $\mathcal{T}_n = \{t_n, t_n + 1, \cdots, t_{n+1} - 1\}, \forall n \in [N-1], \mathcal{T}_N = \{t_N, \cdots, T\}$. When $t = t_n$, the inequality holds. Thus we consider $\forall t \in \mathcal{T}_n \setminus \{t_n\}$, line 9 was triggered, hence $z_{t+1} = z_t = z_{t_n+1}$ and $\|w_t\| \leq z_t$. On the other hand, by property (2) $z_{t_n+1} = 2z_{t_n}$ and $\|w_{t_n}\| > z_{t_n}$. Thus

$$2\|w_{t_n}\| > 2z_{t_n} = z_{t_n+1} = z_t \geq \|w_t\|, \quad \forall t \in \mathcal{T}_n \setminus \{t_n\}$$

(4) since $z_1 = \tau$ and $z_{t+1}$ is either through line 5 (double) or line 9 (maintain). Thus non-decreasing property holds.

Suppose line 5 was never executed, then $z_{T+1} = z_1 = \tau$. Now we consider line 5 was executed at least once. Let $t^* \in [T]$ be the last time step in which line 5 was executed. Thus

$$z_{T+1} = z_T = \cdots = z_{t^*+1} = 2z_{t^*} < 2\|w_{t^*}\|$$

a further upper bound is $z_t \leq 2 \max_t \|w_t\|$ for $t \in [t^* + 1, T + 1]$, combing with $z_t$ being non-decreasing, we complete the proof.

$\square$

# H. Error Correction

We provides the error correction effort as a result of trigger signals $\alpha_t, \beta_t$ from FILTER and tracker, respectively and the chosen regularizer $r_t(w) = f_t(w) + a_t\|w\|^2$ as discussed in Section 6.3. We aim to bound OFFSET := ERROR $-$ CORRECTION by spliting it into two components:

$$\text{OFFSET} = \underbrace{\sum_{t \notin \mathcal{P}} \|g_t - \tilde{g}_t^c\|\|w_t\| - \sum_{t=1}^{T} \alpha_t\|w_t\|^2}_{\text{OFFSET}_1:\text{ due to adaptive clipping}} + \underbrace{\sum_{t \in \mathcal{P}} \|g_t - \tilde{g}_t\|\|w_t\| - \sum_{t=1}^{T} \beta_t\|w_t\|^2 - \sum_{t=1}^{T} f_t(w_t)}_{\text{OFFSET}_2:\text{ due to corruption}}$$

and BIAS:

$$\text{BIAS} = \|u\|^2 \sum_{t=1}^{T} a_t + \sum_t f_t(u) + \|u\| \sum_{t \in \mathcal{P}} \|g_t - \tilde{g}_t^c\| + \|u\| \sum_{t \notin \mathcal{P}} \|g_t - \tilde{g}_t^c\|$$

We begin with a helper Lemma followed by the upper bound.

**Lemma H.1** (Error of Truncated Gradients). *Suppose $g_t, \tilde{g}_t$ satisfies assumptions in Equation (3) and (4). Define $\tilde{g}_t^c$ as in Equation (5) with $h_t \leq \tau$ for some $\tau > 0$. Then*

$$\sum_{t=1}^{T} \|g_t - \tilde{g}_t^c\| \leq 2k \max(\tau, G)$$

*Proof.* By definition: $\|g_t - \tilde{g}_t^c\| \leq G + \tau \leq 2\max(\tau, G)$. Thus

$$\sum_{t=1}^{T} \|g_t - \tilde{g}_t^c\| = \sum_{t=1}^{T} \min\left(\|g_t - \tilde{g}_t^c\|, 2\max(\tau, G)\right)$$

$$\leq 2\max\left(\frac{\tau}{G}, 1\right) \sum_{t=1}^{T} \min\left(\|g_t - \tilde{g}_t^c\|, G\right)$$

$$\leq 2k\max(\tau, G)$$

where the last step is due to Equation (4). $\square$

**Lemma H.2.** *Suppose $g_t, \tilde{g}_t$ satisfies assumptions in Equation (3) and (4). Algorithm 3 and 4 are initialized with some $\tau_G, \tau_D > 0$. Algorithm 3 in response to $\tilde{g}_t$ and output $h_{t+1}$, Algorithm 4 in response to arbitrary sequence $w_t$ in which $\max_t \|w_t\| \le \frac{\epsilon}{2} 2^T$, and outputs $z_{t+1}$. Define*

$$r_t(w) = f_t(w) + a_t \|w\|^2$$

*where $f_t$ is defined as shown in Equation (7) for some $c > 0, p = \ln T, \alpha = \epsilon \tau_G / c, a_t = \alpha_t + \beta_t, \alpha_t, \beta_t$ are defined as in Equation (11) and (12). For some $\gamma_\alpha, \gamma_\beta > 0$:*

$$\text{OFFSET} \le \frac{64 \max(\tau_G, G)^2}{\gamma_\alpha} (k+1) \max\left(\left\lceil \log_2 \frac{8G}{\tau_G} \right\rceil, 1\right)$$

$$+ \epsilon \tau_G + \frac{16k^2 \max(\tau_G, G)^2}{\gamma_\beta} \ln \frac{64k^2 \max(\tau_G, G)^2}{c\gamma_\beta \tau_D} + \frac{c}{2}\tau_D + 2k\tau_D \max(\tau_G, G)$$

*and*

$$\text{BIAS} \le \tilde{O}\left((\gamma_\alpha(k+1) + \gamma_\beta)\|u\|^2 + c\|u\| + \|u\|k \max(\tau_G, G)\right)$$

*Proof.* We show three components in turn:

OFFSET$_1$: due to adaptive clipping:

$$\text{OFFSET}_1 := \sum_{t \notin \mathcal{P}} \|g_t - \tilde{g}_t^c\|\|w_t\| - \alpha_t\|w_t\|^2 \le \sum_{t \notin \mathcal{P}} (G + h_t)\|w_t\| - \alpha_t\|w_t\|^2 \tag{20}$$

For each fixed $t \in \bar{\mathcal{P}}$, we have $A_t\|w_t\| - \alpha_t\|w_t\|^2 \le \sup_{X \ge 0} A_t X - \alpha_t X^2 \le \frac{A_t^2}{4\alpha_t}$, where $A_t = G + h_t > 0$. Hence an upper bound to Equation (20) can be derived by substitute $\alpha_t = \gamma_\alpha, \forall t \in \bar{\mathcal{P}}$:

$$\text{OFFSET}_1 \le \sum_{t \notin \mathcal{P}} \frac{(G + h_t)^2}{4\alpha_t} = \frac{1}{4\gamma_\alpha} \sum_{t \notin \mathcal{P}} (G + h_t)^2 \le \frac{(G + h_T)^2}{4\gamma_\alpha}|\bar{\mathcal{P}}| \le \frac{64 \max(\tau_G, G)^2}{\gamma_\alpha}(k+1)\max\left(\left\lceil \log_2 \frac{8G}{\tau_G} \right\rceil, 1\right)$$

where the last inequality is due to upperbound to $|\bar{\mathcal{P}}|$ by Lemma F.2 (4).

OFFSET$_2$: due to corruption:

The upper bound is obtained through two steps. In each step we aim to show:

$$\text{OFFSET}_2 := \underbrace{\sum_{t \in \mathcal{P}} \|g_t - \tilde{g}_t\|\|w_t\| - \sum_{t=1}^T \beta_t\|w_t\|^2 - \sum_{t=1}^T r_t(w_t)}_{\text{step 1: } \le O(G^2 k \log(\max_t \|w_t\|))} \le \underbrace{O\left(G^2 k \ln(\max_t \|w_t\|)\right) - \sum_{t=1}^T f_t(w_t)}_{\text{step 2: } \le O(G^2 k)}$$

By construction, we have

$$\beta_t = \begin{cases} \beta_t = \gamma_\beta \cdot \frac{\mathbb{1}\{z_{t+1} \ne z_t\}}{1 + \sum_{i=1}^t \mathbb{1}\{z_{i+1} \ne z_i\}}, & t = t_n, n \in [N] \\ 0, & \text{otherwise} \end{cases}$$

Proceed with analysis to step 1, where second line is by Lemma G.1 property (1) and value of $\beta_t$ displayed above:

$$\text{step 1} := \sum_{t \in \mathcal{P}} \|g_t - \tilde{g}_t\|\|w_t\| - \sum_{t=1}^T \beta_t\|w_t\|^2$$

$$= \sum_{n=0}^N \sum_{t \in \mathcal{P} \cap \mathcal{T}_n} \|g_t - \tilde{g}_t\|\|w_t\| - \sum_{n=1}^N \beta_{t_n}\|w_{t_n}\|^2$$

$$\le \sum_{t \in \mathcal{P} \cap \mathcal{T}_0} \|g_t - \tilde{g}_t\|\|w_t\| + \sum_{n=1}^N 2\|w_{t_n}\| \sum_{t \in \mathcal{P} \cap \mathcal{T}_n} \|g_t - \tilde{g}_t\| - \sum_{n=1}^N \beta_{t_n}\|w_{t_n}\|^2$$

$$\le \tau_D \sum_{t \in \mathcal{P} \cap \mathcal{T}_0} \|g_t - \tilde{g}_t\| + \sum_{n=1}^N 2\|w_{t_n}\| \sum_{t \in \mathcal{P} \cap \mathcal{T}_n} \|g_t - \tilde{g}_t\| - \sum_{n=1}^N \beta_{t_n}\|w_{t_n}\|^2$$

where the third line is due to Lemma G.1 property (3). For the first summand, we can define some "imaginary" truncated gradients

$$z_t = \frac{\tilde{g}_t}{\|\tilde{g}_t\|} \min\left(\|\tilde{g}_t\|, \tau_G\right)$$

Notice $\sum_{t \in \mathcal{P} \cap \mathcal{T}_0} \|g_t - \tilde{g}_t\| \leq \sum_{t=1}^{T} \|g_t - z_t\| \leq 2k \max(\tau_G, G)$ by evoking Lemma H.1. Thus,

$$\text{step 1} \leq 2k\tau_D \max(\tau_G, G) + \sum_{n=1}^{N} 2\|w_{t_n}\| \sum_{t \in \mathcal{P} \cap \mathcal{T}_n} \|g_t - \tilde{g}_t\| - \sum_{n=1}^{N} \beta_{t_n} \|w_{t_n}\|^2 \tag{21}$$

Now we analyze each summands over $n$ in Equation (21). Considering a fixed $n \in [N]$:

$$
\begin{aligned}
2\|w_{t_n}\| \sum_{t \in \mathcal{P} \cap \mathcal{T}_n} \|g_t - \tilde{g}_t\| - \beta_{t_n} \|w_{t_n}\|^2 &\leq \sup_{X \geq 0} X \sum_{t \in \mathcal{P} \cap \mathcal{T}_n} 2\|g_t - \tilde{g}_t\| - \beta_{t_n} X^2 \\
&= \frac{\left(\sum_{t \in \mathcal{P} \cap \mathcal{T}_n} \|g_t - \tilde{g}_t\|\right)^2}{\beta_{t_n}} \\
&= \frac{2}{\gamma_\beta} \left(\sum_{t \in \mathcal{P} \cap \mathcal{T}_n} \|g_t - \tilde{g}_t\|\right)^2 \left(1 + \sum_{i=1}^{t} \mathbb{1}\{z_{i+1} \neq z_i\}\right) \\
&\leq \frac{2}{\gamma_\beta} \left(\sum_{t \in \mathcal{P} \cap \mathcal{T}_n} \|g_t - \tilde{g}_t\|\right)^2 \left(1 + \sum_{i=1}^{T} \mathbb{1}\{z_{i+1} \neq z_i\}\right) \\
&= \frac{2}{\gamma_\beta} \left(\sum_{t \in \mathcal{P} \cap \mathcal{T}_n} \|g_t - \tilde{g}_t\|\right)^2 (1 + N)
\end{aligned}
$$

where the second to last line is due to number of $z_{t+1}$ doubled $N = \sum_{t=1}^{T} \mathbb{1}\{z_{i+1} \neq z_i\}$. Now, we substitute it back to equation (21)

$$
\begin{aligned}
\text{step 1} &\leq 2k\tau_D \max(\tau_G, G) + \frac{2(1+N)}{\gamma_\beta} \sum_{n=1}^{N} \left(\sum_{t \in \mathcal{P} \cap \mathcal{T}_n} \|g_t - \tilde{g}_t\|\right)^2 \\
&\leq 2k\tau_D \max(\tau_G, G) + \frac{2(1+N)}{\gamma_\beta} \left(\sum_{n=1}^{N} \sum_{t \in \mathcal{P} \cap \mathcal{T}_n} \|g_t - \tilde{g}_t\|\right)^2 \\
&= 2k\tau_D \max(\tau_G, G) + \frac{2(1+N)}{\gamma_\beta} \left(\sum_{t \in \mathcal{P}} \|g_t - \tilde{g}_t\|\right)^2 \\
&\leq 2k\tau_D \max(\tau_G, G) + \frac{2(1+N)}{\gamma_\beta} \left(\sum_{t=1}^{T} \|g_t - \tilde{g}_t^c\|\right)^2 \\
&\leq 2k\tau_D \max(\tau_G, G) + \frac{2(1+N)}{\gamma_\beta} (2k \max(h_T, G))^2
\end{aligned}
$$

where the last step is due to Lemma H.1 by noticing $\|\tilde{g}_t^c\| \leq h_T, \forall t \in [T]$. This mean we obtained an upper bound to step 1:

$$\text{step 1} := \sum_{t \in \mathcal{P}} \|g_t - \tilde{g}_t\| \|w_t\| - \sum_{t=1}^{T} \beta_t \|w_t\|^2 \leq 2k\tau_D \max(\tau_G, G) + \frac{8k^2 \max{(\tau_G, G)}^2 (1 + N)}{\gamma_\beta}$$

$$\leq 2k\tau_D \max(\tau_G, G) + \frac{8k^2 \max{(\tau_G, G)}^2 \left(1 + \max\left(0, \log_2 \frac{2\max_t \|w_t\|}{\tau_D}\right)\right)}{\gamma_\beta}$$

$$\leq 2k\tau_D \max(\tau_G, G) + \frac{8k^2 \max{(\tau_G, G)}^2 \log_2 \left(2 + \frac{4\max_t \|w_t\|}{\tau_D}\right)}{\gamma_\beta}$$

$$\leq 2k\tau_D \max(\tau_G, G) + \frac{16k^2 \max{(\tau_G, G)}^2 \ln \left(2 + \frac{4\max_t \|w_t\|}{\tau_D}\right)}{\gamma_\beta}$$

where the second step is due to Lemma G.1 (1). Thus, it is sufficient to bound step 2 defined as follows to obtain a bound for $\text{OFFSET}_2$ that is independent of $\max_t \|w_t\|$:

$$\text{step 2} := \frac{16k^2 \max{(\tau_G, G)}^2 \ln \left(2 + \frac{4\max_t \|w_t\|}{\tau_D}\right)}{\gamma_\beta} - \sum_{t=1}^{T} f_t(w_t)$$

evoke Lemma B.1 with $\alpha = \epsilon\tau_G/c$

$$\leq \frac{16k^2 \max{(\tau_G, G)}^2 \ln \left(2 + \frac{4\max_t \|w_t\|}{\tau_D}\right)}{\gamma_\beta} - c\max_t \|w_t\| + \epsilon\tau_G$$

$$\leq \sup_{X > -2} \frac{16k^2 \max{(\tau_G, G)}^2}{\gamma_\beta} \ln(2 + X) - \frac{c\tau_D}{4} X + \epsilon\tau_G$$

for $A, B > 0, A\ln(2 + X) - BX$ obtains its supremum at $X = A/B - 2 > -2$. Hence $\sup_{X > -2} A\ln(2 + X) - BX = A\ln(A/B) - A + 2B$. By substituting $A = \frac{16k^2 \max(\tau_G, G)^2}{\gamma_\beta}, B = \frac{c\tau_D}{4}$ we have

$$= \frac{16k^2 \max{(\tau_G, G)}^2}{\gamma_\beta} \left(\ln \frac{64k^2 \max{(\tau_G, G)}^2}{c\gamma_\beta\tau_D} - 1\right) + \frac{c}{2}\tau_D + \epsilon\tau_G$$

Thus step 1 and step 2 implies

$$\text{OFFSET}_2 \leq \epsilon\tau_G + \frac{16k^2 \max{(\tau_G, G)}^2}{\gamma_\beta} \ln \frac{64k^2 \max{(\tau_G, G)}^2}{c\gamma_\beta\tau_D} + \frac{c}{2}\tau_D + 2k\tau_D \max(\tau_G, G)$$

BIAS: comparator related term

$$\text{BIAS} := \|u\|^2 \sum_{t=1}^{T} a_t + \sum_t f_t(u) + \|u\| \sum_{t \in \mathcal{P}} \|g_t - \tilde{g}_t^c\| + \|u\| \sum_{t \notin \mathcal{P}} \|g_t - \tilde{g}_t^c\|$$

$$= \|u\|^2 \sum_{t=1}^{T} a_t + \sum_t f_t(u) + \|u\| \sum_{t=1}^{T} \|g_t - \tilde{g}_t^c\|$$

$$\leq \|u\|^2 \sum_{t=1}^{T} a_t + \sum_t f_t(u) + \|u\| 2k \max(h_T, G)$$

$$\leq \|u\|^2 \sum_{t=1}^{T} a_t + \sum_t f_t(u) + \|u\| 16k \max(\tau_G, G) \tag{22}$$

where the last It remains to show the first two terms in Equation (22) can be bounded by desired orders. For the first summand, $\sum_t a_t = \sum_t \alpha_t + \sum_t \beta_t$. Thus by definition and Lemma F.2 (4):

$$\sum_t \alpha_t \leq \gamma_\alpha |\mathcal{P}| \leq \gamma_\alpha(k+1) \max\left(\left\lceil \log_2 \frac{8G}{\tau_G} \right\rceil\right) \leq \tilde{O}(\gamma_\alpha(k+1))$$

and Lemma G.1 (1) implies $N \leq O(\ln \max_t \|w_t\|)$ and the condition $\max_t \|w_t\| \leq \frac{\epsilon}{2} 2^T$ implies:

$$\sum_t \beta_t = \gamma_\beta \sum_{t=1}^{T} \frac{\mathbb{1}\{z_{t+1} \neq z_t\}}{1 + \sum_{i=1}^{t} \mathbb{1}\{z_{i+1} \neq z_i\}} = \gamma_\beta \ln(1+N) \leq \tilde{O}(\gamma_\beta)$$

The second term in Equation (22) can be upper bounded by Lemma B.1 by substituting $\alpha = \epsilon \tau_G / c$:

$$\sum_{t=1}^{T} f_t(u) \leq 3c \ln T \|u\| \left[ \ln\left(1 + \left(\frac{\|u\|}{\alpha}\right)^{\ln T}\right) + 2 \right] = \tilde{O}(c\|u\|)$$

Thus,

$$\text{BIAS} \leq \tilde{O}\left((\gamma_\alpha(k+1) + \gamma_\beta)\|u\|^2 + c\|u\| + \|u\|k\max(\tau_G, G)\right)$$

$\square$

# I. Base Algorithms for unknown G

In this section, we show the *Epigraph-based regularization* scheme developed by (Cutkosky & Mhammedi, 2024) guarantees appropriate composite regret defined as:

$$R_T^{\mathcal{A}}(u) := \sum_{t=1}^{T} \langle g_t, w_t - u \rangle + r_t(w_t) - r_t(u)$$

when $r_t(w) = f_t(w) + a_t\|w\|^2$ and the sequence $a_t$ satisfies assumption in Lemma H.2. That is

$$R_T^{\mathcal{A}}(u) := \sum_{t=1}^{T} \langle g_t, w_t - u \rangle + f_t(w_t) - f_t(u) + \sum_{t=1}^{T} a_t\left(\|w_t\|^2 - \|u\|^2\right)$$

Thus, can be used as a base algorithm $\mathcal{A}$ that can be supplied to Algorithm 1 when the $G \geq \max_t \|g_t\|$ is unknown. The proof is identical to (Cutkosky & Mhammedi, 2024) except $a_t$ behaves differently here. Nevertheless, we summarize as Algorithm 5 and shows the regret guarantee attained stilled suffices in achieving our aim.

*Epigraph-based Regularization* is a geometric reparameterization: it suffices to design two learners: $\mathcal{A}_w$ to $w_t$ and $\mathcal{A}_y$ to produce $y_t$, where $(w_t, y_t) \in W = \{(w, y) : y \geq \|w\|^2\} \subseteq \mathbb{R}^{d+1}$. Then it suffices to design algorithm to bound:

$$R_T^{\mathcal{A}}(u) \leq \underbrace{\sum_{t=1}^{T} \langle g_t, w_t - u \rangle + f_t(w_t) - f_t(u)}_{R_T^{\mathcal{A}_w}(u)} + \underbrace{\sum_{t=1}^{T} a_t(y_t - \|u\|^2)}_{R_T^{\mathcal{A}_y}(u)} \tag{23}$$

This is a sum of two regrets for the pair $w_t$ and $y_t$ subject to $y_t \geq \|w_t\|^2$. This problem can be solved by using a pair of unconstrained learners $(\mathcal{A}_w, \mathcal{A}_y)$ that produce $(\hat{w}_t, \hat{y}_t) \in \mathbb{R}^{d+1}$ and guarantee regret bounds:

$$\tilde{R}_T^{\mathcal{A}_w}(u) \geq \sum_{t=1}^{T} \langle g_t, \hat{w}_t - u \rangle + f_t(\hat{w}_t) - f_t(u)$$

and

$$\tilde{R}_T^{\mathcal{A}_y}(u) \geq \sum_{t=1}^{T} a_t(\hat{y}_t - \|u\|^2)$$

By a black-box conversion from unconstrained-to-constrained learning due to (Cutkosky & Orabona, 2018) (Theorem 3) to enforce the constraint: this involves a projection $\Pi_W : \mathbb{R}^{d+1} \to W := \{(w, y) : y \geq \|w\|^2\}$ (line 7) and a certain technical correction to the gradient feedback (11-15). These procedures output variables $(w_t, y_t) = \Pi_W((\hat{w}_t, \hat{y}_t))$ that satisfies the constraint and guarantees $R_T^{\mathcal{A}_w}(u) \leq \tilde{R}_T^{\mathcal{A}_w}(u)$ and $R_T^{\mathcal{A}_y}(u) \leq \tilde{R}_T^{\mathcal{A}_y}(u)$. In particular, we choose Algorithm 2 as $\mathcal{A}^w$ and (Jacobsen & Cutkosky, 2022) Algorithm 4 as $\mathcal{A}^y$. We summarize the entire procedure as Algorithm 5 and its gaurantee on $R_T^{\mathcal{A}}(u)$ is displayed as Theorem I.3. We first present a useful definition and a helper Lemma.

**Definition I.1.** For the set $W = \{(w, y) : y \geq \|w\|^2\} \subseteq \mathbb{R}^{d+1}$, and arbitrary $(w, y) \in W$ and $(\hat{w}, \hat{y}) \in \mathbb{R}^{d+1}$ and some $h_t, \gamma > 0$:

(1) norm: $\|(w, y)\|_t = h_t^2 \|w\|^2 + \gamma^2 y^2$

(2) dual norm: $\|(w, y)\|_{*,t} = \frac{\|w\|^2}{h_t^2} + \frac{y^2}{\gamma^2}$

(3) distance function of $(\hat{w}, \hat{y})$ to $W$: $S_t((\hat{w}, \hat{y})) = \inf_{y \geq \|w\|^2} \|(w, y) - (\hat{w}, \hat{y})\|_t$

(4) subgradient at $(\hat{w}, \hat{y})$: $\nabla S_t((\hat{w}, \hat{y})) = \left( \frac{h_t^2(\hat{w}-w)}{h_t^2\|\hat{w}-w\|^2+\gamma^2(\hat{y}-y)^2}, \frac{\gamma^2(\hat{y}-y)}{h_t^2\|\hat{w}-w\|^2+\gamma^2(\hat{y}-y)^2} \right)$

(5) projection map $\Pi_W^t((\hat{w}, \hat{y})) = \operatorname{argmin}_{(w,y) \in W} \|(w, y) - (\hat{w}, \hat{y})\|_t$

**Lemma I.2.** *In the same notation as Definition I.1, if $\|g_t\| \leq h_t$ and $\alpha_t \in [0, \gamma]$, and $(\delta_t^w, \delta_t^y) = \|(g_t, a_t)\|_{*,t} \nabla S_t((\hat{w}_t, \hat{y}_t))$ then*

$$\|\delta_t^w\| \leq \sqrt{2}h_t, \qquad |\delta_t^y| \leq \sqrt{2}\gamma$$

*Proof.* Since $\|g_t\| \leq h_t$ and $\alpha_t \in [0, \gamma]$, $\|(g_t, a_t)\|_{*,t} \leq 2$. On the other hand $\|\nabla S_t((\hat{w}, \hat{y}))\|_{*,t} = 1$, and

$$\|(\delta_t^w, \delta_t^y)\|_{*,t} = \frac{\|\delta_t^w\|^2}{h_t^2} + \frac{|\delta_t^y|^2}{\gamma^2}$$

Thus

$$\frac{\|\delta_t^w\|^2}{h_t^2} + \frac{|\delta_t^y|^2}{\gamma^2} \leq 2$$

This implies both $\frac{\|\delta_t^w\|^2}{h_t^2} \leq 2$ and $\frac{|\delta_t^y|^2}{\gamma^2} \leq 2$. $\qquad\square$

**Theorem I.3.** *Suppose $g_t, \tilde{g}_t$ satisfies assumptions in Equation (3) and (4), and having access to $\tilde{g}_t^c$ as defined in Equation (5) with $h_t$ provided by* FILTER *(Algorithm 3). with $\alpha = \epsilon/c, \gamma = \gamma_\alpha + \gamma_\beta$, for some $\epsilon, c, \gamma_\alpha, \gamma_\beta, \tau_G, \tau_D > 0$, Algorithm 5 guarantees:*

$$R_T^{\mathcal{A}}(u) \leq \tilde{O}\left( \epsilon h_T + \|u\| h_T \sqrt{T} + \|u\|^2 (\gamma_\alpha(k+1) + \gamma_\beta) \right)$$

*In addition, the produced iterate satisfies $\max_t \|w_t\| \leq \frac{\epsilon}{2} 2^T$*

*Proof.* Notice that we used $\hat{w}_t, \hat{y}_t$ as outputs from some unconstrained learner and $w_t, y_t$ being their projection on $W_t$ in Algorithm 5 denote. We begin our analysis from Equation (23):

$$R_T^{\mathcal{A}}(u) \leq \sum_{t=1}^{T} \langle g_t, w_t - u \rangle + f_t(w_t) - f_t(u) + \sum_{t=1}^{T} a_t(y_t - \|u\|^2)$$

By Cutkosky & Orabona (2018) Theorem 3

$$\leq \sum_{t=1}^{T} \langle \tilde{g}_t^c + \delta_t^w, \hat{w}_t - u \rangle + f_t(w_t) - f_t(u) + \sum_{t=1}^{T} (a_t + \delta_t^y)(y_t - \|u\|^2)$$

Notice $\|\hat{w}_t\| \leq \|w_t\|$, thus $f_t(w_t) \leq f_t(\hat{w}_t)$

$$\leq \underbrace{\sum_{t=1}^{T} \langle \tilde{g}_t^c + \delta_t^w, \hat{w}_t - u \rangle + f_t(\hat{w}_t) - f_t(u)}_{R_T^{\mathcal{A}_w}(u)} + \underbrace{\sum_{t=1}^{T} (a_t + \delta_t^y)(y_t - \|u\|^2)}_{R_T^{\mathcal{A}_y}(u)} \tag{24}$$

Since $\gamma_\beta = \frac{\gamma}{2}$, $a_t = \alpha_t + \beta_t \leq \gamma_\alpha + \gamma_\beta = \gamma$. Thus, by Lemma I.2, $\|\tilde{g}_t^c + \delta_t^w\| \leq h_t + \sqrt{2}h_t \leq 3h_t$ and $|a_t + \delta_t^y| \leq \gamma + \sqrt{2}\gamma \leq 3\gamma$. By Theorem C.2:

$$R_T^{\mathcal{A}^w}(u) \leq \tilde{O}\left( \epsilon h_T + \|u\| h_T \sqrt{T} \right)$$

and Theorem 10 of Cutkosky & Mhammedi (2024) shown

$$R_T^{\mathcal{A}_y}(u) \leq \tilde{O}\left( \epsilon h_T + \|u\|^2 \sqrt{\gamma^2 + \gamma \sum_{t=1}^{T} a_t} \right)$$

Thus, we can bound Equation (24) by combine both bounds:

$$R_T^{\mathcal{A}}(u) \leq \tilde{O}\left( \epsilon h_T + \|u\| h_T \sqrt{T} + \|u\|^2 \sqrt{\gamma^2 + \gamma \sum_{t=1}^{T} a_t} \right)$$

since $\sum_t a_t = \sum_t \alpha_t + \sum_t \beta_t$, by the same computation as that of Lemma H.2, $\sum_t \alpha_t \leq \tilde{O}(\gamma_\alpha(k+1))$ and $\sum_t \beta_t \leq O(\gamma_\beta)$

$$\leq \tilde{O}\left( \epsilon h_T + \|u\| h_T \sqrt{T} + \|u\|^2 \sqrt{\gamma^2 + \gamma\left((\gamma_\alpha(k+1) + \gamma_\beta)\right)} \right)$$

$\gamma \leq \gamma_\alpha$ and $\gamma \leq \gamma_\beta$

$$\leq \tilde{O}\left( \epsilon h_T + \|u\| h_T \sqrt{T} + \|u\|^2 \left(\gamma_\alpha(k+1) + \gamma_\beta\right) \right)$$

$\square$

---

**Algorithm 5** Robust Online Learning By Exploiting In-Time Offset

---

1: **Input:** Time horizon $T$, FILTER (Algorithm 3), TRACKER (Algorithm 4),
   an algorithm $\mathcal{A}_y$ with optimal rate in parameter-free literature (e.g., Jacobsen & Cutkosky (2022) Algorithm 4),
   corruption parameter $k$, base algorithm parameters $\epsilon$, regularization parameters $c, \alpha, \gamma_\alpha, \gamma_\beta, \gamma$.
2: **Initialize:** Initialize Algorithm 2 as $\mathcal{A}_w$ with $\epsilon$. Initialize $\mathcal{A}_y$ with $\epsilon$.
   Initialize FILTER with $\tau_G$ (outputs $h_t$ as a conservative lower-bound guess for $G$).
   Initialize TRACKER with $\tau_D$ (outputs $z_t$ as a conservative lower-bound guess for $\max_t |w_t|$).
3: **for** $t = 1$ **to** $T$ **do**
4:     Receive $\hat{w}_t$ from $\mathcal{A}_w$; Receive $\hat{y}_t$ from $\mathcal{A}_y$.
5:     Compute operators in Definition I.1 with $h_t, \gamma$.
6:     *# Explicit projection of $(\hat{w}_t, \hat{y}_t)$ through projection map $\Pi_W^t$ as in Definition I.1.*
7:     Compute projection: $(w_t, y_t) = \Pi_W^t((\hat{w}_t, \hat{y}_t))$.
8:     Play $w_t$, receive $\tilde{g}_t^c, h_{t+1}$ from FILTER. Send $w_t$ to TRACKER and receive $z_{t+1}$.
9:     Compute $\alpha_t, \beta_t$ as defined in Equations (11), (12).
10:    Compute quadratic regularizer weights: $a_t = \alpha_t + \beta_t$.
11:    *# Compute gradient correction direction $(\delta_t^w, \delta_t^y)$ with $\|\cdot\|_{*,t}$ and $\nabla S_t$ as in Definition I.1.*
12:    Compute: $(\delta_t^w, \delta_t^y) = \|(\tilde{g}_t^c, a_t)\|_{*,t} \nabla S_t((\hat{w}_t, \hat{y}_t))$.
13:    *# Send corrected gradients:*
14:    Send $\left(\frac{1}{2}\left(\tilde{g}_t^c + \delta_t^w\right), 2h_{t+1}\right)$ to $\mathcal{A}_w$.
15:    Send $\frac{1}{2}(a_t + \delta_t^y)$ and $\frac{3}{2}\gamma$ to $\mathcal{A}_y$.
16: **end for**

---

## J. Regret Analysis with Unknown $G$

We provide the regret guarantee of Algorithm 1 when using an instance of Algorithm 5 as a base learner $\mathcal{A}$.

**Theorem J.1** (Restated Theorem 6.1). *Suppose $g_t, \tilde{g}_t$ satisfies assumptions in Equation (3) and (4). Setting $r_t(w) = f_t(w) + \alpha_t\|w\|^2$, where $f_t$ is defined in Equation (7) with parameters: $\alpha = \epsilon\tau_G/c, p = \ln T, \gamma = \gamma_\alpha + \gamma_\beta$, for some $\epsilon, c, \gamma_\alpha, \gamma_\beta, \tau_G, \tau_D > 0$, there exists an algorithm $\mathcal{A}$ as a base algorithm such that Algorithm 1 guarantees:*

$$R_T(u) \leq \tilde{O}\left(8\epsilon \max(\tau_G, G) + \|u\| \max(\tau_G, G)\left(\sqrt{T} + k\right) + c\|u\| + (\gamma_\alpha(k+1) + \gamma_\beta)\|u\|^2\right)$$

$$+ \frac{16k^2 \max(\tau_G, G)^2}{\gamma_\beta} \ln \frac{64k^2 \max(\tau_G, G)^2}{c\gamma_\beta \tau_D} + \frac{c}{2}\tau_D + 2k\tau_D \max(\tau_G, G)$$

$$+ \frac{64(k+1)\max(\tau_G, G)^2}{\gamma_\alpha} \max\left(\left\lceil \log_2 \frac{8G}{\tau_G}\right\rceil, 1\right)$$

*Proof.* Note the definition of $R_T(u)$, the proof is completed by combining Lemma H.2 and Theorem I.3 $\qquad \square$

## K. Fenchel Conjugate

Here we collects basic properties of *Fenchel conjugate*, see reference such as (Bertsekas, 2009; Orabona, 2019), and previously established Lemma used in Appendix E for completeness.

**Definition K.1.** Let $f : \mathbb{R}^d \to [-\infty, \infty]$, the *Fenchel conjugate* $f^*$ is defined as

$$f^*(\theta) = \sup_{x \in \mathbb{R}^d} \langle \theta, x \rangle - f(x)$$

the *double conjugate* $f^{**}$ is defined as

$$f^{**}(\theta) = \sup_{x \in \mathbb{R}^d} \langle \theta, x \rangle - f^*(x)$$

**Theorem K.2.** *Let $f : \mathbb{R}^d \to (-\infty, \infty]$*

1. *$f(x) = f^{**}(x), \forall x \in \mathbb{R}^d$ iff $f$ is convex and lower semicontinuous*

2. $\langle \theta, x \rangle - f(x)$ *achieves its supremum in $x$ at $x = x^*$ iff $x^* \in \nabla f^*(\theta)$*

**Lemma K.3.** *(Theorem A.32 of (Orabona, 2019)) The Lambert function $W : \mathbb{R}^+ \to \mathbb{R}^+$ is defined as*

$$x = W(x) \exp(W(x)), \quad \text{for } x > 0$$

*and $W(x) > 0.6 \ln(1 + x)$ for $x > 0$.*

**Lemma K.4.** *(Theorem A.3 of (Orabona, 2019)) Let $a, b > 0$, $f(x) = b \exp(x^2/2a)$. Then the Fenchel conjugate is*

$$f^*(\theta) = \sqrt{a}|\theta| \left( \sqrt{W\left(\frac{a\theta^2}{b^2}\right)} - \frac{1}{\sqrt{W\left(\frac{a\theta^2}{b^2}\right)}} \right)$$

*where $W(\cdot)$ is the Lambert function.*

**Lemma K.5.**

$$\sqrt{x} - \frac{1}{\sqrt{x}} \geq \sqrt{\frac{x}{9}}, \quad \forall x \geq \frac{3}{2}$$

*Proof.* The proof is based on rearrange $x \geq \frac{3}{2}$, the condition is equivalent to

$$\left(1 - \frac{1}{3}\right) x \geq 1$$

Given $x > 0$, divide both side by $\sqrt{x}$

$$\left(1 - \frac{1}{3}\right) \sqrt{x} \geq \frac{1}{\sqrt{x}}$$

Rearrange and we complete the proof. $\square$

