# OpenReview forum: "Unconstrained Robust Online Convex Optimization"
_ICML.cc/2025/Conference — ICML 2025 poster_

### Official Review · Reviewer_h1f3 · 2025-03-11

**Overall Recommendation:** 4

**Summary:**

The paper presents an algorithm to solve unconstrained OCO when the observed gradient might be corrupted. They first present an algorithm when $G := \max_t ||g_t||$ is known, by truncating the observed gradient to ensure the norm is less than a specified $h_t$, and adding a regularizer to limit the growth of $||w_t||$. When using $h_t = G$, this results in a guarantee in $O(||u|| G (\sqrt{T} + k)$ where $k$ measures the amount of corruption.

When $G$ is not known, the authors use a clever doubling trick to ensure that in each epoch to ensure that $h_t$ is close to  $\max_t ||w_t||$. They also add a quadratic regularizer when doubling to ensure that the error from truncation remains bounded. This result in a similar regret guarantee with an additional $O(||u||^2 + G^2) k$.

## update after rebuttal

I decided to maintain my score.

**Claims And Evidence:**

All claims seems to be thoroughly proved.

**Essential References Not Discussed:**

None that I can think of.

**Experimental Designs Or Analyses:**

N/A

**Methods And Evaluation Criteria:**

The criteria used is the regret, which is standard in online learning.

**Other Comments Or Suggestions:**

Typos:
- line 20, abstract: $||u|| \to u$.
- line 69 $1\{True\} = 0 \to 1
- Multiple times $|u| \to ||u||$ or $|w_t| \to ||w_t||$.
- line 697: unclear what terms are in the $\sum_{t=1}^T$, maybe add a parenthesis.

**Other Strengths And Weaknesses:**

Strength:
1- It is the very first approach to the unconstrained robust OCO in the literature
2- They combine well several tricks, from unconstrained OCO and robust OCO, to solve it. They also provide new ones that arise from this unique setting

Weaknesses:

**Questions For Authors:**

Your algorithm focus on $|| ||$ being the euclidean norm but could it be applied for other norms with their associated dual norm?

**Relation To Broader Scientific Literature:**

As far as I know, this is the first paper to tackle OCO with corrupted gradients in the unconstrained setting.
- van Erven et al., 2021 tackled the case of potential outliers in the standard constrained setting, which is a special case of this paper.

**Theoretical Claims:**

The authors explain the ideas behind each of the theorems. I read mainly the ideas detailed in the main body of the paper, but not the proofs in appendix.

---

> ### Author Rebuttal · Authors · 2025-04-01
>
> We appreciate the positive feedback. Regarding norms, our algorithm extends to other norm settings as long as $k$ is measured accordingly and dual norms. We will also revise the manuscript to fix typos.

---

### Official Review · Reviewer_EewU · 2025-03-15

**Overall Recommendation:** 4

**Summary:**

The paper addresses the case of online learning on the unconstrained domain with corrupted gradient feedback with no assumptions on the corruptions nature. The paper provides an algorithm with regret guarantee $\|u\|G(\sqrt{T} + k)$ for the case when the
Lipschitz constant is known, and provide the algorithm extension with adaptive thresholding with an extra additive penalty of $(\|u\|^2 + G^2)k$ where $k$ reflects the total corruption level, and $u$ is any comparison point. The paper provides very thorough introduction, discusses all the challenges of the considered problem setup, carefully provides the analysis of both cases of known and unknown Lipschitz constant giving a lot of intuition and clear flow.

## Update after rebuttal:

I thank the authors for the response. I have also read the other reviews, and I decided to keep my initial score.

**Claims And Evidence:**

Yes. However, it should be considered that in the non-differentiable case the sub-gradients might be non-unique.

**Essential References Not Discussed:**

Not that I am aware of.

**Experimental Designs Or Analyses:**

Not applicable

**Methods And Evaluation Criteria:**

Yes

**Other Comments Or Suggestions:**

- In line 198, right side, I believe that the sum of $r_t$ should be lower bounded not by the $O(\cdot)$ but rather $\Omega$ notation since it is a lower bound.  Just a rate in $O$ will not be enough, even if $r_t$ grows with the same rate as error, but slower 2 times, their subtraction with not be cancelled out but growing.
- line 178 right side: Upper bound on ERROR I believe should be $2kG \max_t\|\omega_t\|$ instead of $kG \max_t\|\omega_t\|$. That will influence also on the constant $c$ later in line 218 right side and further.

**Other Strengths And Weaknesses:**

The paper is very clearly written and is a pleasure to read. All statements seem to be thorough, relations to the previous related works are discussed.

**Questions For Authors:**

- line 331: If we want to bound a truncation error, $E_{\bar P}$ defined in eq. (10) as $\|g_t - \tilde g_t^c\|$, why here do we bound $\|\tilde g_t - \tilde g_t^c\|$?

**Relation To Broader Scientific Literature:**

They provide new regret bounds for the unconstrained online convex learning with corrupted feedback. The work in certain sense extends to the setting on the unconstrained OCO and improves upon a method in van Erven et al. (2021) which considered bounded domain. The previous works on unconstrained OCO only considered stochastic unbiased feedback perturbations.

**Theoretical Claims:**

Yes, just the ones in the main body of the paper, I did not check the appendix. There is an unclear part in line 331, I wrote a question regarding it in the questions section.

---

> ### Author Rebuttal · Authors · 2025-04-01
>
> We thank the reviewer for the positive feedback. On the non-differentiable case: we agree that OCO via linearized losses naturally extends to subgradients, and thus our results apply to non-differentiable convex functions as well.
>
> Regarding the term $E_{\bar P}$ (line 331, left), this should indeed be $g_t - \tilde{g}_t^c$ as suggested. This was a typo in the main text; the expression was handled correctly in Appendix (line 1282) and was referred to as $offset_1$. We will revise the manuscript accordingly

---

### Official Review · Reviewer_Z9rr · 2025-03-16

**Overall Recommendation:** 3

**Summary:**

The authors investigate online convex optimization (OCO) in an unconstrained domain under corrupted gradient feedback. They introduce a new measure of corruption, denoted as $k$, which accounts for both the number of corrupted rounds and the magnitude of gradient deviations. Given $k$, their proposed algorithm achieves a regret bound of $\mathcal{O}(\| u \| G (\sqrt{T} + k))$ for any comparator, provided the gradient norm bound for uncorrupted rounds is known. In cases where the gradient norm bound is unknown, they propose a filtering approach that guarantees a similar regret bound with an additional term of $\mathcal{O}( (\| u \|^2 + G^2) k)$. Additionally, they establish matching lower bounds (up to logarithmic factors) for any choice of comparator $u$.

## update after rebuttal
I would like to thank the authors for their response, and I have decided to retain my original positive score.

**Claims And Evidence:**

The claims are theoretical and are supported by thorough discussions and arguments presented in the main text, which appear sound and reasonable. However, I have not reviewed the proofs in the supplementary material in detail.

**Essential References Not Discussed:**

The references are cited and discussed adequately.

**Experimental Designs Or Analyses:**

The paper only has theoretical contributions.

**Methods And Evaluation Criteria:**

The paper only has theoretical contributions.

**Other Comments Or Suggestions:**

A few minor typos:

1. Line 187, col 2: \cite should be replaced with \citep.
2. Line 330, col 1: "the a" appear together

**Other Strengths And Weaknesses:**

In my view, the paper’s primary strength lies in the development of a filtering technique to address the unknown gradient norm bound and associated challenges in an unbounded domain. The paper is well-written, providing a strong motivation for the problem while clearly articulating both the problem itself and the main obstacles in addressing it. The authors effectively discuss their techniques and results, making the work accessible and easy to understand.

The primary weakness of the paper is its reliance on established ideas, especially their similarities to Zhang & Cutkosky (2022), which diminishes the overall novelty of the work. The key technical tools employed are well-known, which somewhat limits the scope of the contributions.

**Questions For Authors:**

The paragraph before Corollary 6.3 claims that the regret with respect to the baseline point $u = 0$ is **constant** no matter what $k$ is. However, the regret bound in Corollary 6.3 has a $\mathcal{O}(k)$ dependency. Is there something that I am missing here?

**Relation To Broader Scientific Literature:**

The paper borrows some ideas from Zhang & Cutkosky (2022), Cutkosky & Mhammedi (2024), and van Erven et al. (2021). In particular, it adopts the composite loss function method from Zhang & Cutkosky (2022) to handle large corrupted gradients. The authors propose a novel filtering technique to handle unknown bounds on the gradient norms of uncorrupted rounds, which serves a similar purpose to the filtering technique used in van Erven et al. (2021).

**Theoretical Claims:**

The discussions and arguments presented in the main text appear sound and reasonable. However, I have not reviewed the proofs in the supplementary material in detail.

---

> ### Author Rebuttal · Authors · 2025-04-01
>
> We thank the reviewer for validating the theoretical contributions. Regarding the constant regret interpretation after Corollary 6.2, it relies on setting $\tau_D = O(1/k)$, which is an initialization parameter used in the doubling trick to track $\max_{i \le t} |w_t|$. Note that if $k$ is unknown, we may set $\tau_D=1/\sqrt{T}$ to achieve constant Regret(0) for and $k\le \sqrt{T}$, which the range in which our algorithm suffers $\tilde O(\sqrt{T})$ regret overall. Apologies for not specifying this properly in the theorem statement - we will clarify this in the manuscript and correct the noted typos.

---

### Official Review · Reviewer_w7uA · 2025-03-17

**Overall Recommendation:** 3

**Summary:**

The paper studies the challenging problem of online convex optimization (OCO) in an unconstrained domain under the presence of adversarially corrupted gradient feedback. Unlike classical OCO, where gradient estimates are assumed to be accurate or only subject to benign noise, this work makes no statistical assumptions on the corruptions. Instead, it considers arbitrary (and potentially adversarial) deviations in the gradient signals. A regret bound is provided in the paper. The matching lower bounds are provided that demonstrate the tightness of the upper bounds under certain regimes.

## update after rebuttal

I appreciate the authors for the rebuttal. Although it does not address my concerns, I found some insight for the parameter setting issue. I decide to keep my original score.

**Claims And Evidence:**

The claims of the paper are clear and have evidence.

**Essential References Not Discussed:**

NA.

**Experimental Designs Or Analyses:**

1. The numerical study of the proposed algorithm is limited. Including more experiments on simulation data and real datasets that illustrate the algorithm’s performance under various corruption regimes could provide practical validation of the theoretical findings.
 2. Parameter Analysis: A sensitivity analysis regarding the choice of hyperparameters would be a welcome addition, potentially guiding practitioners in implementing the methods.

**Methods And Evaluation Criteria:**

The theoretical parts of the paper is clear, but I think this paper lacks some convincing numerical studies to support the theoretical findings.

**Other Comments Or Suggestions:**

Please see the comments above. Some numerical evidences are highly recommended to be added.

**Other Strengths And Weaknesses:**

Please see the above comments.

**Questions For Authors:**

Please see the comments above.

**Relation To Broader Scientific Literature:**

The theoretical findings of this paper are interesting which can contribute to the area of online optimization algorithms.

**Theoretical Claims:**

While the theoretical contributions of this paper is interesting, the paper would benefit from a more extensive discussion on the practical implications of the algorithms. For example, insights into how the proposed methods perform on real data or in simulated adversarial settings would strengthen the overall impact.

Although the paper analyzed the regret bounds under more general assumptions, the bound is still built based on an implicit assumption that the gradient is bounded, which may not be true for some real applications (or the upper-bound is very large).

---

> ### Author Rebuttal · Authors · 2025-04-01
>
> We thank the reviewer for the constructive feedback. As this work focuses on the theoretical foundations of robust online convex optimization, we look forward to systematically investigate practical applications in future work.
>
> In terms of hyper-parameter, the only required user input is the corruption level $k$. Even without exact knowledge of $k$, our theoretical results suggest setting $k = O(\sqrt{T})$ guarantees the classical OCO regret bound $\tilde{O}(\sqrt{T})$ as long as the true corruption level is less than $\sqrt{T}$, which tolerates a significant range of corruption problems. We also look forward to an empirical study of the impact of mis-specifying $k$.
>
> In terms of the scenario of gradient-norm being large, existing theory in adversarial online learning suggests regret scales with the maximum gradient norm. Our results actually help combat this somewhat: a few large outlier gradients could be modeled as “corruptions”, and our regret would not scale with their value. Nevertheless, we acknowledge this limitation and are interested in future developments that mitigate this when large gradients are common.

---

### Decision · Program_Chairs · 2025-05-01

**Decision:**

Accept (poster)

**Comment:**

This paper studies online convex optimization in an unconstrained setting with adversarially corrupted gradient feedback. The authors propose algorithms with provable regret guarantees for both known and unknown gradient bounds, using a novel filtering strategy and tailored regularization to maintain robustness. The work addresses a challenging and underexplored problem and provides tight theoretical guarantees with clear motivation and analysis.

The paper is well-written and makes a solid theoretical contribution. Strengths include the originality of the problem setting, the soundness of the proposed methods, and the clear structure of the presentation. The main weakness is the lack of empirical validation, though the authors position this as future work. Minor issues around notation and clarity were acknowledged and addressed.